# Recent Advances of Stretchable Nanomaterial-Based Hydrogels for Wearable Sensors and Electrophysiological Signals Monitoring

**DOI:** 10.3390/nano14171398

**Published:** 2024-08-27

**Authors:** Haiyang Duan, Yilong Zhang, Yitao Zhang, Pengcheng Zhu, Yanchao Mao

**Affiliations:** Key Laboratory of Materials Physics of Ministry of Education, School of Physics, Zhengzhou University, Zhengzhou 450001, China; dhy777@gs.zzu.edu.cn (H.D.); zyl0816@stu.zzu.edu.cn (Y.Z.); yitaozhang@gs.zzu.edu.cn (Y.Z.)

**Keywords:** nanomaterials, hydrogel, wearable sensor, electrophysiological signals monitoring

## Abstract

Electrophysiological monitoring is a commonly used medical procedure designed to capture the electrical signals generated by the body and promptly identify any abnormal health conditions. Wearable sensors are of great significance in signal acquisition for electrophysiological monitoring. Traditional electrophysiological monitoring devices are often bulky and have many complex accessories and thus, are only suitable for limited application scenarios. Hydrogels optimized based on nanomaterials are lightweight with excellent stretchable and electrical properties, solving the problem of high-quality signal acquisition for wearable sensors. Therefore, the development of hydrogels based on nanomaterials brings tremendous potential for wearable physiological signal monitoring sensors. This review first introduces the latest advancement of hydrogels made from different nanomaterials, such as nanocarbon materials, nanometal materials, and two-dimensional transition metal compounds, in physiological signal monitoring sensors. Second, the versatile properties of these stretchable composite hydrogel sensors are reviewed. Then, their applications in various electrophysiological signal monitoring, such as electrocardiogram monitoring, electromyographic signal analysis, and electroencephalogram monitoring, are discussed. Finally, the current application status and future development prospects of nanomaterial-optimized hydrogels in wearable physiological signal monitoring sensors are summarized. We hope this review will inspire future development of wearable electrophysiological signal monitoring sensors using nanomaterial-based hydrogels.

## 1. Introduction

In recent years, wearable sensors have rapidly emerged in the field of physiological signal monitoring, becoming an alternative solution for traditional rigid and bulky sensors [1,2]. These novel wearable sensors play significant roles in various areas, from medical monitoring, such as electrocardiography (ECG), electroencephalography (EEG), and electromyography (EMG) [3,4,5,6], to health management, such as activity tracking. In addition, wearable sensors are widely used in non-contact control and virtual reality (VR), permeating into every corner of life [7]. However, the traditional wearable devices used for monitoring physiological signals face numerous issues during usage [8,9]. Firstly, traditional monitoring devices still lack adequate flexibility, making it difficult to synchronize with the soft human skin and tissues during movement, resulting in difficulties with capturing subtle movements, thus affecting monitoring effectiveness [10,11,12,13]. Secondly, these devices often lack inherent adhesion and require external force for fixation, leading to skin discomfort and poor biocompatibility [14,15]. Moreover, traditional wearable physiological monitoring devices can only be used for skin surface monitoring and cannot be implanted inside the body [3,16,17]. In addition, the sensitivity and response time of traditional wearable devices still need to be improved to meet the demand for high-standard medical diagnoses in emergency situations [18,19,20]. Therefore, the developing of novel wearable sensors with excellent comprehensive performance to achieve accurate, convenient, and continuous monitoring of physiological signals is imperative.

In 1960, Professors Wichterle and Lim prepared the first generation of hydrogels, which are three-dimensional network polymer materials containing a large amount of water [21]. With the development of materials science, hydrogels doped with nanomaterials have made a great leap in all aspects compared to the first generation, making them the preferred material for preparing excellent wearable electrophysiological sensors [22,23,24,25,26]. Firstly, nanomaterial-based hydrogels possess excellent electrical properties, enabling the prepared wearable sensors to detect subtle changes caused by tiny deformations, thus exhibiting outstanding sensitivity and a rapid response time [27,28,29]. Secondly, nanomaterial composite hydrogels exhibit excellent mechanical properties, which is one of the important advantages in making wearable sensors [30,31,32]. Their outstanding stretchability makes the sensor less prone to damage; the lower modulus enables hydrogel sensors to adhere better to human skin and organ tissues, ensuring good conformal contact and obtaining high-quality signals. Additionally, nanomaterial-based hydrogels also have good adhesion [33,34,35], which is important for wearable sensors to prevent unexpected peeling off from the human body. Based on these advantages, the nanomaterial-based hydrogel has shown broad prospects in monitoring electrophysiological signals.

In this review, we discuss the recent research advances in nanomaterial-based hydrogel wearable sensors for electrophysiological signal monitoring from three perspectives, illustrated in Figure 1. In the first part, the classes of nanomaterials used to construct nanomaterial composite hydrogel for wearable sensors are discussed, including carbon nanomaterials, metal nanomaterials, and two-dimensional transition metal nanomaterials. In the second part, we summarize the main properties of hydrogel-based wearable sensors, including the electrical properties, adhesion properties, and mechanical properties. Next, the applications of hydrogel-based wearable sensors for electrophysiological signal monitoring are summarized, such as ECG, EMG, as well as EEG signals detection. Finally, a conclusion and outlook are given to the future development trend based on the current hydrogel-based wearable sensors, aiming at realizing a wider range of application scenarios for hydrogel-based electrophysiological monitoring sensors.

## 2. Nanomaterials for Preparing Hydrogels

Hydrogels are flexible materials with excellent comprehensive properties and have been widely applied in various fields such as medical monitoring, artificial intelligence, and rehabilitation training [42]. With the development of nanomaterial science, one or more types of nanomaterials have been introduced into the hydrogel networks to improve their functional properties [43]. The addition of nanomaterials significantly improves the electrical conductivity, structural stability, and detection sensitivity of hydrogel materials [44]. Therefore, the integration of hydrogel with wearable sensors has broad application prospects in physiological signal monitoring.

Compared with hydrogels without nanomaterial doping, the addition of nanomaterials can significantly improve the conductivity of hydrogels [45]. This is mainly because some nanomaterials themselves have superior conductive properties, uniformly incorporating conductive nanomaterials into the hydrogel network, which can significantly enhance the conductivity [46]. Additionally, after adding nanomaterials, extensive conductive channels can be formed inside the hydrogels, which facilitates ion exchange within the hydrogels, further enhancing their conductivity [47]. On the other hand, these nanomaterial-doped hydrogels typically exhibit excellent mechanical properties, capable of withstanding high levels of tension, compression, and torsion, along with characteristics such as high toughness, fatigue resistance, and self-healing [48]. The self-healing capability of hydrogels is often attributed to their dynamic crosslinking network. After over-stretching, dynamic bonds within hydrogels would break, followed by forming new dynamic crosslinks in the breaking interface to be self-healed. Additionally, certain functional groups on the surface of nanomaterials, as well as the hydrogen bonds between hydrogel molecular chains, play a role in self-healing by providing energy dissipation and strength. The self-healing properties of hydrogels confer numerous benefits, including extended service life and enhanced stability and reliability in harsh environments. Moreover, nanomaterial-doped hydrogels often demonstrate high stability under some special environments, maintaining structural stability and unaffected conductivity, even underwater. These unique properties endow nanomaterial-based hydrogels with broader application prospects, playing an important role in wearable sensing fields such as medical care, remote monitoring, and non-contact control in electrophysiological monitoring [49,50,51]. 

The field of nanomaterials research encompasses a variety of high-performance materials, with carbon-based nanomaterials standing out prominently, including graphene and carbon nanotubes [17,28]. On the other hand, attention is focused on metal nanowires, liquid metals, as well as metal nanoparticles [10,47], each exhibiting unique physicochemical properties. Meanwhile, two-dimensional metal nanomaterials such as MXene [8,9] are garnering significant interest due to their distinctive structures and properties. It is noteworthy that hydrogel materials prepared through the chelation of metal ions with ligands have also demonstrated exceptional performance characteristics [52,53], holding potential applications across multiple fields. Table 1 summarizes the different properties of various nanomaterial-doped stretchable hydrogel-based wearable sensors for physiological signal monitoring.

### 2.1. Carbon Nanomaterials

In the preparation of stretchable nanomaterial-based hydrogels used in physiological signal monitoring, carbon nanomaterials have been widely applied [64,65,66,67,68]. Firstly, carbon nanomaterials possess inherent high conductivity, promoting the transport of electrons within the hydrogel, thereby providing excellent conductivity for the hydrogel [69,70,71]. Secondly, carbon nanomaterials exhibit various morphologies and structures. Among them, the layered structure can trigger quantum tunneling effects, imparting the hydrogel with the ability to detect extremely small strains, playing an important role in the application of wearable sensor devices [72,73,74]. Additionally, the incorporation of carbon nanomaterials enables the hydrogel to achieve higher sensitivity when subjected to external forces, providing a basis for the real-time monitoring of physiological signals in wearable devices [75,76,77]. Some carbon nanomaterials, such as carbon nanotubes (CNTs) and graphene, have an ultra-high aspect ratio and can construct hydrogel networks through physical interactions. CNTs’ tubular structure enables π-π stacking with polymer chains in the hydrogel, whereas graphene’s large specific surface area facilitates surface functionalization, allowing for the formation of hydrogen bonds with the hydrogel. These interfacial interactions effectively promote adhesion and binding between the materials, contributing to the formation of a stable network structure, endowing the hydrogel with excellent mechanical properties, and providing an important application foundation for wearable sensors [46,78,79]. Adhesion is also a crucial property for wearable sensors. Surface modification of carbon nanomaterials could also increase the adhesion between the composite hydrogel and tissue interface, making it better suited for wearable sensor applications [80,81,82]. Carbon nanomaterials used for doping in hydrogels applied in wearable sensors for physiological signal monitoring usually include graphene and carbon nanotubes (CNTs).

Graphene, with its unique layered structure, surface modifiability, good breathability, and outstanding conductivity, has become an ideal nanomaterial for preparing wearable sensors [48,83]. It plays a crucial role in fields such as flexible electronics, biomedicine, and smart sensors [84,85]. Lu et al. [86] reported an eco-friendly graphene-doped hydrogel with good biocompatibility, biodegradability, and mechanical properties. As shown in Figure 2a, the hydrogel is synthesized by the interaction between starch and polyvinyl alcohol (PVA), forming a physical crosslinking network through numerous hydrogen bonds. The incorporated graphene surfaces are rich in oxygen-containing groups, which can easily interact with carboxyl groups on the hydrogel network. By adding customized ionic liquids, the uniform distribution of graphene nanomaterials in the hydrogel network is further promoted. These graphene-based interactions impart the hydrogel with a robust structure, good stability, mechanical strength, and increased conductivity. Figure 2b exhibits the scanning electron microscope (SEM) images, showing the detailed internal structure of the hydrogel and the distribution of graphene nanofillers. It can be seen that the network structure formed by starch and PVA is very rough, but the introduced graphene fills the gaps of the rough structure, resulting in a tighter internal network structure of the hydrogel. Furthermore, with the introduction of ionic liquids (ILs), the filling effect of graphene becomes more significant, and the hydrogel exhibits a smoother surface and a more uniformly compact internal structure. This uniform and compact internal structure is crucial for the stability and high mechanical strength of the hydrogel.

Carbon nanotubes, due to their biocompatibility, excellent conductivity, and surface modifiability, are ideal materials for the fabrication of composite hydrogels [87,88,89]. Hydrogels doped with carbon nanotubes can be used in the development of wearable sensors, smart electronic skins [90,91], and electric stimulation therapy, among other applications [92,93]. Seo et al. [94] designed and fabricated a hydrogel capable of maintaining functionality in the human body for an extended period of time. As shown in Figure 2c, the natural compound, tannic acid (TA), containing abundant catechol and galloyl functional groups, was added as a crosslinker to polyvinyl alcohol (PVA) hydrogel, where abundant hydrogen bonds provide outstanding mechanical strength to the hydrogel. Afterward, by introducing carboxyl and hydroxyl functionalized carbon nanotubes (fCNTs), fCNTs can be more uniformly dispersed in the hydrogel, endowing the hydrogel to achieve excellent conductivity. In Figure 2d, the transmission electron microscope (TEM) image of the hydrogel clearly shows its nanomorphology, demonstrating the uniform distribution of fCNTs in the hydrogel and the morphology characteristics of fCNTs. This fCNTs-composite hydrogel exhibits excellent mechanical properties, including a modulus range (10~100 kPa) matched with tissues, high toughness (400–873 J/m^3^), high elongation (≈1000% strain), rapid self-healing capability (within 5 min), and high conductivity (≈40 S/m). A wearable smart sensor was then developed by using the fCNT nanomaterial-based hydrogels. The sensor demonstrates excellent conductivity and sensitivity to detect slight deformations, which can monitor subtle physiological activities in daily life scenarios. Additionally, the excellent mechanical properties of the hydrogel enable great durability, which is crucial for wearable smart sensors used in long-term physiological signal monitoring.

Hydrogels doped with carbon nanotubes for the preparation of wearable sensors not only have high electrical conductivity but also possess a modulus perfectly matched to the skin and organs [95,96,97]. Zhang et al. [98] reported a porous ion-conductive hydrogel prepared from polyacrylamide and carbon nanotubes with a modulus of 80 kPa, completely matching the skin and organs (such as the heart). Furthermore, this all-hydrogel battery has been applied in wearable devices due to its high stability and biocompatibility. As shown in Figure 2e, the anode and cathode of this battery were made of active hydrogels doped with carbon nanotubes (CNTs). The original hydrogel was first dehydrated into a dehydrated electrode, followed by contact with the hydrogel electrolyte, which completed the rehydration process. After rehydration, the dehydrated electrode absorbed moisture and softened again, forming a fused interface with the electrolyte, thereby achieving the structural and performance stability of the all-hydrogel battery. In Figure 2f, the dehydrated hydrogel skeleton structure is displayed. It exhibits a typical macroporous sponge-like morphology with an extremely low modulus (12 kPa), but its conductivity is relatively low at this stage. It is found that there is a trade-off relationship between the conductivity and modulus of the hydrogel. Therefore, by adjusting the water absorption rate during the rehydration process, the hydrogel eventually achieves a modulus (70 kPa) matched to biological tissues and high conductivity (103 mS/cm). This research designed a type of biocompatible hydrogel with both a low modulus and high conductivity, which provides guidance for developing power sources for wearable electronics with mechanical properties matched to tissues.

**Figure 2 nanomaterials-14-01398-f002:**
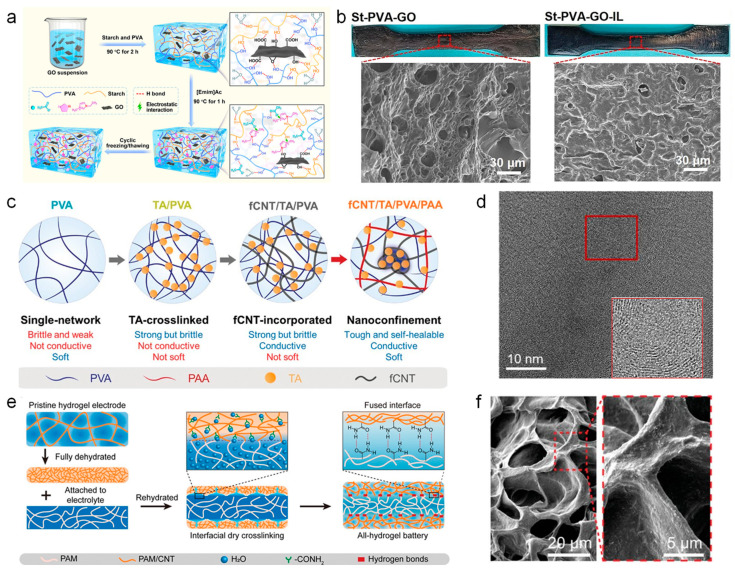
Hydrogel-based wearable sensors prepared from carbon nanomaterials. (**a**,**b**) Eco-friendly, highly stretchable, and biodegradable graphene-doped ionic hydrogel-based wearable sensors for daily human motion monitoring [86]. (**c**,**d**) Carbon nanotube-doped stretchable hydrogel-based wearable sensors for underwater electromyography signal monitoring [94]. (**e**,**f**) Low-modulus ultra-soft hydrogel-based wearable devices doped with carbon nanotubes for powering wearable electronics [98].

### 2.2. Nano Metal Materials

Compared to carbon nanomaterials, metal nanomaterials have become another excellent candidate for preparing nano-material-based hydrogels due to their unique advantages such as excellent conductivity, low cost, low toxicity, and outstanding rheological properties [99]. These outstanding characteristics indicate significant potential for developing the next generation of wearable smart sensor devices [100]. Typically, metal nanomaterials come in various forms, including metal nanowires, liquid metal, and metal nanoparticles [101,102,103]. In recent years, these hydrogels doped with metal nanomaterials have attracted increasing research attention for applications in smart wearable sensors [104]. Thus far, metal nanomaterials doped in hydrogel-based wearable sensors include zero-dimensional metal nanoparticles, one-dimensional metal nanowires, and fluid-like liquid metals [105,106,107].

Metal nanowires such as silver nanowires (AgNWs) exhibit excellent conductivity and mechanical properties that can be adjusted by changing the crosslink density while also providing hydrogels with excellent antibacterial activity, making them an ideal choice for hydrogel fabrication [108]. Inspired by the layered structure of the skin, Chen et al. [109] reported an encapsulated hydrogel strain sensor composed of injectable hydrogel conductive wires and an outer hydrogel film capable of monitoring the electrophysiological signals of human movement. Figure 3a illustrates the fabrication process and schematic diagram of the hydrogel doped with silver nanowires (AgNWs). AgNWs were firstly injected into alginate (AA)-derived self-healing hydrogels to form AgNWs/AA conductive hydrogels with a filamentous structure. Subsequently, the AgNWs/AA hydrogel filaments were embedded into methyl acrylate alginate (AA-MA) hydrogel films through an in situ photopolymerization process. The introduction of AgNWs imparts excellent mechanical and electrical properties to the hydrogel. The hydrogel exhibits excellent tensile strength (22.6 ± 1.3 kPa), with a Young’s modulus (15.6 ± 1.0 kPa) perfectly matched to the skin. The strain sensor prepared by the hydrogel shows good electrical properties, including sensitivity (GF = 1.63), as well as a short relaxation time (166 ms). Figure 3b shows the scanning electron microscope (SEM) images of the hydrogel microstructure, which clearly reveals a dense network structure of the AgNWs on the hydrogel surface, forming a second conductive network of entangled AgNWs within the hydrogel. When directly adhered to the fingers, elbows, knees, and throat, the composite hydrogel-based strain sensor can directly reflect changes in human movements through electrical signals. Additionally, the sensor can be further used for monitoring electrophysiological signals, making it applicable in areas such as health monitoring, gesture recognition, and medical diagnostics.

The unique rheological properties of liquid metals provide significant advantages over metal nanowires, and this property has led to excellent suitability and efficacy of liquid metals in hydrogel doping applications. Liquid metal refers to metallic elements or alloys that remain in a liquid state at room temperature. As a metal-based nanomaterial that is doped into hydrogels, liquid metal exhibits good flowability, low costs, excellent biocompatibility, as well as outstanding thermal and electrical conductivity, thereby holding significant potential for applications in health care [110], sensor technology, aerospace, etc. Yeo et al. [111] successfully introduced a particle model of liquid metals into cellulose hydrogel matrices (LMCs) containing liquid metal particle models by introducing gallium (Ga)-based liquid metal particles (LMPs) into a cellulose hydrogel matrix, which enabled the hydrogel to accurately monitor physiological signals due to an enhanced mechanical responsiveness and electrochemical performance. As depicted in Figure 3c, the incorporation of Ga ions into the hydrogel led to additional ion crosslinking, reinforcing the hydrogel framework and increasing its modulus by 18 times. Additionally, the LMCs demonstrate remarkable mechanical stability, which can endure up to 80% compression and still maintain structural integrity after 20 cycles of compression. These outstanding mechanical properties endow the hydrogel with the potential to prepare durable wearable sensors. Apart from their excellent mechanical performance, the incorporation of LMPs also imparts high electrical properties to the hydrogel, resulting in a lower impedance (1.5 × 10^4^ ohms). Furthermore, due to the presence of LMPs, the hydrogel’s resistance can recover to its original state after repeated compressions, indicating good potential for the preparation of wearable sensors. Figure 3d displays SEM images of the LMC hydrogel samples after freeze-drying, which confirms the enhancement of hydrogel crosslinking by doping Ga ions. It can be observed that the internal structure of the hydrogel features thick cellulose frameworks and distinct LMPs. The thicker cellulose framework indicates the Ga ion promotion effect of ion crosslinking, leading to the formation of thicker cellulose frameworks. The introduction of liquid metals significantly improves the structure of the hydrogel skeleton, giving it excellent fracture resistance and higher stress and deformation tolerance, which together contribute to the excellent mechanical properties of hydrogels.

Metal nanoparticles refer to metallic particles with sizes ranging from 1 to 100 nanometers, possessing excellent antibacterial properties, anti-inflammatory capabilities, conductivity, and a large specific surface area, endowing them with broad potentials as nanofillers to regulate hydrogel properties [112]. Zhang et al. [113] proposed a biomimetic strategy inspired by the skin, adopting in situ reduction to fabricate a tough, robust, antibacterial, and conductive hydrogel. As shown in Figure 3e, in the first step, the cellulose chain self-assembled in ethanol to generate a porous cellulose framework, mimicking collagen fiber in the skin. Subsequently, the in situ production and fixation of silver nanoparticles on the cellulose framework were achieved through the cellulose’s oxidation-reduction reaction with silver nitrate. Finally, a silver nanoparticle-doped polyacrylamide hydrogel sample was obtained through the thermal-initiated polymerization of acrylamide. Figure 3f shows the SEM images of the cellulose framework before and after the reduction of silver nitrate. It can be clearly seen that compared to the white porous cellulose framework, a light yellow and dense surface is observed after the reduction reaction. Meanwhile, silver nanoparticles are uniformly distributed in the hydrogel network. The obtained hydrogel exhibits excellent mechanical properties with a tensile strength reaching 2.0 MPa and a high toughness of 12 MJ/m^3^. It was found that the content of silver nanoparticles was directly related to the mechanical properties of the hydrogel, which could be used for the preparation of wearable sensors with adjustable mechanical properties. Additionally, the developed wearable sensor based on the hydrogel exhibits high electrical conductivity (0.74 S/m) and sensitivity (GF = 4.4), as well as good antibacterial and freeze-resistant properties. These properties further expand its application prospects in wearable sensing devices.

**Figure 3 nanomaterials-14-01398-f003:**
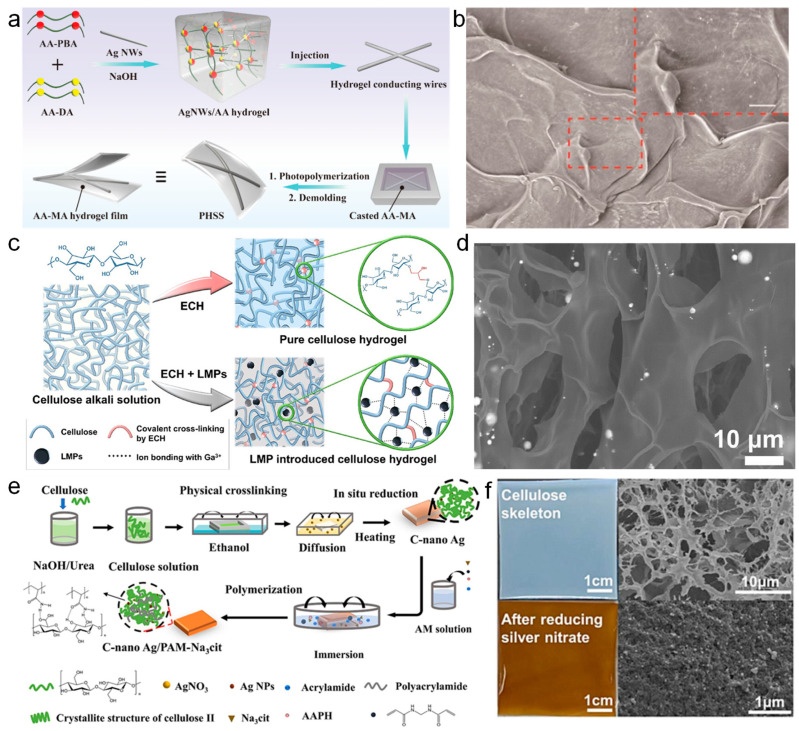
Hydrogel-based wearable sensors prepared from metallic nanomaterials. (**a**,**b**) Silver nanowire-doped hydrogels for wearable strain sensors for physiological signal monitoring [109]. (**c**,**d**) High-performance hydrogels prepared from liquid metal Ga for wearable sensors for glucose detection [111]. (**e**,**f**) Silver nanoparticle hydrogels for wearable sensors with antimicrobial properties [113].

### 2.3. MXene Nanomaterials

Two-dimensional transition metal nanomaterials are another special type of nanomaterial that has been used to construct nanomaterial-based hydrogels with some unique characteristics [114]. Compared with the former two materials, two-dimensional transition metal nanosheets have advantages such as hydrophilic surfaces with negative charges, a large specific surface area, high conductivity, adjustable mechanical properties, and good surface modifiability [115,116]. Therefore, it has aroused great attention in the preparation of conductive hydrogels for wearable sensors, soft robots, medical devices, and energy storage [18,117]. MXene is a common type of two-dimensional transition metal nanomaterial. MXene material exhibits a typical layered structure, with the layers interconnected by van der Waals forces. There are abundant functional groups, such as hydroxyl and carboxyl groups, on their surfaces. In addition, by adjusting the composition of MXene, MXene materials with different physical and chemical properties can be obtained. There has been extensive research and considerable progress in preparing MXene-based hydrogels for wearable sensors [118,119,120].

MXene is a novel type of two-dimensional transition metal nanomaterial with outstanding mechanical strength and high conductivity. Its unique layered structure and abundant surface functional groups [121], as well as its hydrophilicity, make it an ideal active material for the next generation of wearable sensors. Adler et al. [122] prepared a high-performance, stable MXene-based composite hydrogel. The obtained organic hydrogel exhibits high stretchability (>500%), low hysteresis (<3%), excellent fatigue resistance, good adhesion, long-term stability (>7 days), and freeze resistance (−40 °C). Figure 4a shows the synthesis mechanism of this MXene-based hydrogel. The MXene nanosheets were encapsulated by natural antioxidant tannic acid (TA), forming a dense hydrogen bonding network that effectively enhances the stretchability, toughness, and anti-hysteresis of the hydrogel. Meanwhile, the addition of glycine/ammonium water binary solvent ensures the long-term performance stability of the hydrogel under prolonged use and extreme environments. These excellent comprehensive performances benefit from the introduction of MXene nanomaterials, making the composite hydrogel an important material for preparing wearable sensors. Figure 4b shows the SEM images of a hydrogel sample stored for 120 days, with the inset showing the surface morphology of a fresh hydrogel sample. It can be seen from the inset that the surface of the hydrogel samples presents a smooth sheet-like structure, mainly due to the influence of MXene’s layered structure. After storing for 120 days, the surface of the hydrogel sample still retains the smooth nanosheet morphology, providing direct evidence for the excellent water retention ability of the hydrogel. The water retention ability could be attributed to the action of TA, which can effectively protect the structure of MXene nanosheets during long-term storage.

MXene also possesses great surface modifiability, which can be modified by adding functional groups on the surface, enabling MXene to act as a crosslinking agent in the hydrogel system for enhancing various properties of the hydrogel-based wearable sensors. Han et al. [123] utilized MXene to create P(AA-co-AM)/MXene@PDADMAC semi-interpenetrating network (IPN) hydrogels with high electrical conductivity by pre-adsorbing the cationic polymer PDADMAC and by in situ polymerizing anionic monomer tannic acid (AM) and neutral monomer acrylic acid (AA). The fabrication process of this hydrogel is illustrated in Figure 4c. By using MXene@PDADMAC (PDADMAC refers to poly(diallyldimethylammonium chloride)) as a template and common acrylic acid (AA) and acrylamide (AM) as monomers, P(AA-co-AM)/MXene@PDADMAC semi-interpenetrating hydrogels were prepared. Due to the good hydrophilicity of MXene nanosheets and the hydrogen bonds, as well as electrostatic interactions between MXene nanosheets and the polymer network, the hydrogel exhibits excellent mechanical properties, high conductivity, and good water retention ability. The flexible strain sensor based on the hydrogel demonstrates excellent sensitivity (GF = 0.98), with no significant performance degradation after 30 days or 1100 stretching cycles. SEM images of the freeze-dried MXene-gel are shown in Figure 4d; it can be observed that after complete dehydration, the network structure of the hydrogel does not produce any transparent porous structure, as previously reported. Instead, it generates a honeycomb-like cavity structure, which could provide better mechanical properties. Additionally, the honeycomb structure can facilitate a better distribution of the MXene, thereby the electrical conductivity of the hydrogel can be further improved due to the more complete filling of MXene. The study of MXene as a crosslinking agent could provide seminal ideas for the development of novel conductive hydrogels and the application of hydrogel wearable sensors in human motion monitoring.

MXene, as an excellent hydrogel nanofiller, not only provides excellent electrical properties but also brings excellent mechanical properties to the hydrogel, especially for the enhancement of the resilience of the hydrogel [124]. Huang et al. [125] reported a good resilience hydrogel based on hydrophilic and hydrophobic interactions, which has been successfully applied in wearable strain sensors with a wide range of strain sensing capabilities. Figure 4e illustrates the synthesis process of this hydrogel, which is composed of gelatin, hydrophilic acrylamide, hydrophobic octadecyl methacrylate (C18), sodium dodecyl sulfate (SDS), NaCl, and MXene-based crosslinked copolymers. These components are polymerized at room temperature using ammonium persulfate and N,N′-methylenebisacrylamide as the initiator and chemical crosslinking agent, respectively. The resulting hydrogel strain sensor exhibits excellent comprehensive performance, including high stretchability (1224%) and high toughness (560 kJ m^−3^). The hydrogel exhibits exceptional resilience, maintaining its tensile strength after 200 cycles of loading and unloading at 100% strain and fully recovering its original state even at 80% strain, indicating its superior elastic properties and resistance to fatigue under both tensile and compressive loads. Figure 4f shows the cross-sectional SEM images of the hydrogel samples, with the inset displaying a more microscopic view of the hydrogel cross-section. It can be seen that the hydrogel exhibits a “wave-like” wrinkled shape, which is attributed to the incorporation of MXene nanomaterials. Meanwhile, the cross-section of the hydrogel exhibits an orderly distributed internal network structure, which provides more pathways for electron conduction, enabling the hydrogel to possess more stable and durable electrical performance. This study promotes the development of the fabrication of wearable sensing devices with higher sensitivity and broader sensing ranges.

In general, carbon nanomaterials, such as carbon nanotubes and graphene, significantly enhance the mechanical and electrical properties of hydrogels due to their high electrical conductivity and robust network structures. However, achieving a uniform dispersion of these materials during the preparation process remains a significant challenge, as a non-uniform dispersion can diminish their performance. Nanometal materials, including metal nanoparticles and metal nanowires, enhance the conductivity of hydrogels but may excessively increase the modulus, leading to a mismatch of modulus with human tissue. The multifunctionality of two-dimensional transition nanomaterials like MXene is attributed to their surface tunability, yet their fabrication process is usually complex and costly. These varieties of nanofillers can be chosen according to their properties in some aspects for the specific application scenarios.

**Figure 4 nanomaterials-14-01398-f004:**
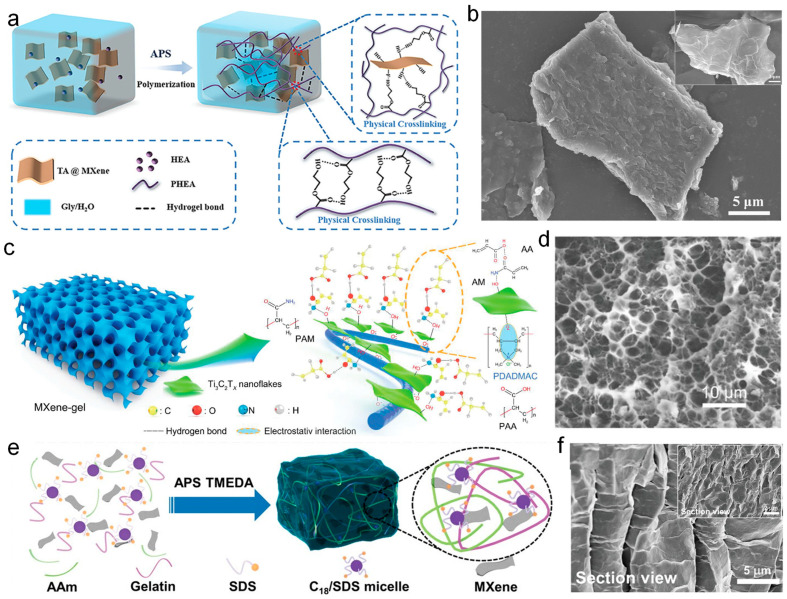
2D transition metal material-prepared hydrogel-based wearable sensor. (**a**,**b**) TA-encapsulated MXene hydrogel-based wearable sensor for monitoring ECG signals [122]. (**c**,**d**) MXene hydrogel-based wearable device for sign language conversion [123]. (**e**,**f**) MXene hydrogel-based wearable sensors for underwater real-time communication and early warning [125].

## 3. Properties of Composite Hydrogel Wearable Sensor

Hydrogel for wearable sensors should have several key properties to ensure its adaptability and functionality. First, its flexibility and stretchability allow it to conform to the curves and movements of the human body [126,127], adapting to body parts of different sizes and shapes, thereby providing a comfortable experience for the user and alleviating discomfort during prolonged wear [128,129]. The hydrogel’s mechanical adjustability allows it to be tailored to the needs of different application scenarios [130]. Additionally, its high conductivity and sensitivity allow it to conduct current efficiently and respond quickly to changes in external stimuli [131,132,133,134]. The hydrogel’s strong adhesion ensures that it binds tightly to different types of surfaces [135,136], providing a stable working environment as it is firmly anchored to the skin or other object surfaces [137]. At the same time, stability is also demonstrated by the ability to resist the effects of various chemicals and different acids and bases in the body’s tissue fluids, maintaining adhesion properties and ensuring the reliable operation of the sensor under a variety of extreme conditions [138]. The water absorption rate of hydrogels significantly influences their mechanical, electrical, and adhesive properties. A higher water absorption rate indicates a greater water content, facilitating the mobility of polymer chains and thereby enhancing flexibility and mechanical robustness, which is beneficial for tissue conformance and sensor applications. Additionally, an increased water absorption rate enhances ionic mobility, which in turn boosts ionic conductivity and electrical performance, leading to an improved sensitivity of wearable sensors. However, excessive water absorption can result in swelling, potentially damaging the structure of the hydrogel sensor. The comprehensive performance of hydrogel materials makes it an important candidate in the field of wearable technology, helping to promote the development and application of smart wearable devices.

### 3.1. Conductivity

Hydrogel-based wearable sensors are crucial devices that are capable of monitoring subtle electrical signals generated by the human body [139,140]. Their unique functionalities, such as high sensitivity and wide sensing range, are enabled by the hydrogel materials they employ, which exhibit good electrical conductivity and stability. To achieve superior conductivity in hydrogels, extensive studies have been conducted and various methods have been proposed. These include doping with conductive nanomaterials such as carbon nanotubes and metal nanoparticles to increase the hydrogel’s conductivity [141,142], improving the fabrication process to enhance the electronic transmission efficiency of hydrogels, and surface modification to facilitate their contact with electrodes, etc. The continuous progress and innovation of these techniques enable hydrogels to meet the requirements of wearable sensors better, achieving an efficient acquisition and transmission of electrophysiological signals from the human body.

By optimizing the conductivity of hydrogels, wearable sensors can monitor and record the electrophysiological status of the human body more accurately, therefore providing more reliable solutions for medical monitoring, health management, and sports tracking [143,144]. The continuous progress and innovation of these technologies have broadened the application prospects of hydrogel-based wearable sensors in fields such as healthcare, wellness, and sports. Kim et al. [145] prepared a highly conductive hydrogel by adding a certain metal ion into the solution, as shown in Figure 5a. Zeolitic imidazolate framework-8 (ZIF-8), as a nanofiller, was combined with poly(acrylamide-co-hydroxyethyl acrylate) (PAAm-co-HEA) copolymer to prepare a highly flexible, stretchable, and durable hydrogel. Adding ZIF-8 nanocrystals and LiCl to the hydrogel matrix significantly increased the ionic conductivity and freeze resistance. Figure 5b demonstrates that with the addition of LiCl, the ionic conductivity of the hydrogel can be significantly improved from 0.1 S/m to 0.68 S/m. Meanwhile, the addition of ZIF-8 enhances the comprehensive performance of the hydrogel sample while maintaining the electrical conductivity at a high level of 0.45 S/m. Such excellent values could fully meet the requirements of wearable sensors for recording physiological electrical signals. Figure 5c intuitively demonstrates the excellent conductivity through the hydrogel-based circuit. It can be seen that the LED light was brightly lit in its original state, and the LED gradually dimmed but still maintained good conductivity, even when stretched to 250%. After the hydrogel returned to its original length, the LED light also returned to its original brightness. This phenomenon illustrates the excellent electrical conductivity and stability of the hydrogel, demonstrating its outstanding potential in wearable sensing devices.

In addition to directly adding ions to the hydrogel, the incorporation of custom-designed ionic liquids is also an alternative strategy to optimize the conductivity of the hydrogel system. Zhang et al. [146] proposed a novel ion-conductive hydrogel by introducing a multifunctional phenylboronic acid ionic liquid (PBA-IL) to the hydrogel matrix. Figure 5d illustrates the fabrication process of this hydrogel. Firstly, the phenylboronic acid ionic liquid PBA-IL was prepared by alkylation reaction, in which the synergistic and multifunctional properties with other groups were successfully realized through the clever design of PBA-IL. Next, PBL-IL and cellulose nanofibrils (CNF) were mixed at room temperature, followed by one-step polymerization under the action of a crosslinking agent; thus, PAM/PBA-IL/CNF hydrogels with semi-interpenetrating networks were prepared. Specifically, the conductive PBA-IL and the carboxyl groups on the CNF surface synergistically promote ion migration, resulting in excellent conductivity. Figure 5e shows the variation in the conductivity of the hydrogel samples prepared with different ionic liquid contents. It can be observed that with the increase in ionic liquid content, the conductivity of the system shows a continuously increasing trend, from an initial ionic conductivity of 0.92 ± 0.10 mS/cm to 6.94 ± 0.21 mS/cm. Compared with the hydrogel sample without ionic liquids, the conductivity increased by more than 7.5 times. This is because the carboxyl groups on CNFs promote the mobility of counterion, thereby enhancing ion movement in the hydrogel network. Considering the synergistic effect of PBA-IL and CNF in enhancing ion conduction, the combination of CNF dynamic crosslinked three-dimensional networks can generate multiple conduction pathways, further facilitating ion transfer and achieving excellent conductivity. Figure 5f clearly demonstrates the outstanding conductivity of the obtained hydrogel samples. The brightness of the LED connected to the hydrogel circuit gradually decreased as the strain increased from 0 to 900%. It can be visually observed that even after stretching to a strain of 900%, the LED still maintained lighting, indicating that the conductive network of the hydrogel could remain stable, even under a large tension. This stable conductivity of hydrogels under extreme strains is a crucial property for the application of wearable sensors.

When considering the design and application of hydrogel-based wearable sensors, in addition to meeting the functional requirements in daily scenarios, the stability of their electrical conductivity should also be fully considered in complex conditions or extreme environments [147]. This requires comprehensive optimizations in various aspects, such as materials, structure, and fabrication processes. Wei et al. [148] developed a swelling-resistant ion-conductive hydrogel and successfully applied it to underwater strain sensing. As shown in Figure 5g, this hydrogel is composed of polyvinyl alcohol (PVA), [2-(methacryloyloxy) ethyl]dimethyl-(3-sulfopropyl) ammonium hydroxide (SBMA), and 2-hydroxyethyl methacrylate (HEMA) copolymers. The addition of hydrochloric acid (HCl) to the mixture induced the protonation of SBMA, which enhanced the hydrogen bonding and ionic interactions within the polymer chains. After freezing/thawing treatment, the mechanical strength of the hydrogel is further increased. Hydrochloric acid (HCl) protonates the negatively charged SO_3_^−^ groups of SBMA, forming a cationic polyelectrolyte hydrogel. Electrostatic repulsion between cations reduces osmotic pressure and decreases swelling, enabling stable performance of the hydrogel, even in extreme conditions such as underwater. Figure 5h demonstrates that with an increasing HCl content, the ion conductivity of the hydrogel samples shows a continuously increasing trend, reaching up to 4.58 S/m. The positively charged R_3_N^+^ groups and negatively charged SO_3_^−^ groups within the copolymer hydrogel form orderly arranged side chains, which serve as ion migration channels under external electric fields. With the introduction of hydrochloric acid, the efficiency of ion migration is greatly improved due to more free ions. Figure 5i illustrates the potential application of the hydrogel sensor in swimming or diving monitoring. By attaching the hydrogel sensor to different parts of the body, real-time motion signals can be monitored during swimming. These results demonstrate the significant potential of this hydrogel in complex conditions such as underwater environments.

**Figure 5 nanomaterials-14-01398-f005:**
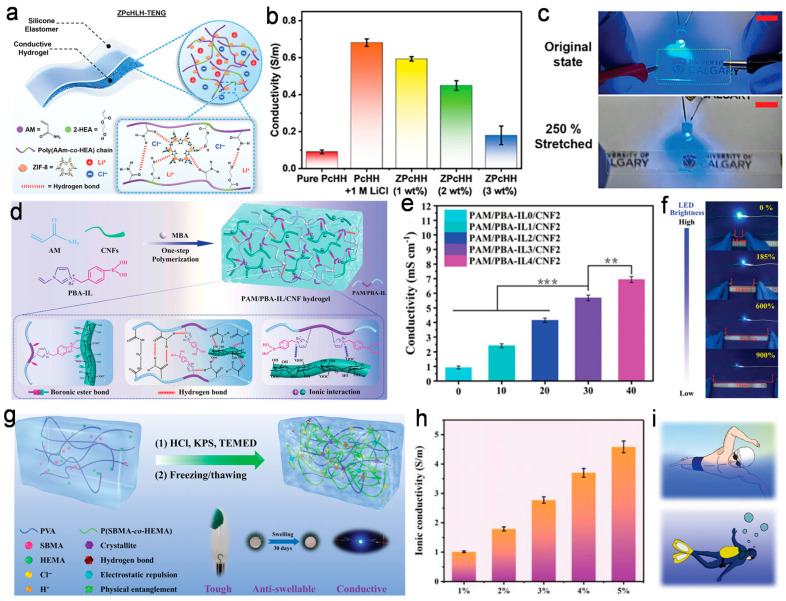
Electrical conductivity of stretchable hydrogel-based wearable sensors. (**a**–**c**) Enhanced electrical conductivity of hydrogel-based wearable sensors by introducing ZIF-8 nanocrystals [145]. (**d**–**f**) Enhanced electrical conductivity of hydrogel-based wearable sensors by introducing PBA-IL ionic liquid [146]. ** *p* < 0.01, *** *p* < 0.001. (**g**–**i**) Enhancement of electrical conductivity of hydrogel-based wearable sensors by introducing SBMA [148].

### 3.2. Adhesion Performance

The outstanding electrical performance of hydrogels endows them with the capability to perceive subtle strains and electrophysiological signals, serving as a crucial foundation for wearable sensors to monitor physiological signals. In addition to the aforementioned characteristics, excellent adhesive properties are vital for the practical applications of wearable sensors. Good adhesive properties ensure that the hydrogels can adhere properly to biological interfaces, thereby decreasing the contact impedance and enhancing the quality and stability of monitoring signals [149]. Furthermore, good adhesion can also provide convenience for the comfortable wearing and versatile application scenarios of wearable sensors. In areas where adhesives are in contact with the skin, there is an increased risk of adverse reactions such as allergies or inflammation. Consequently, ensuring the biocompatibility and safety of the adhesive interface is a paramount consideration that must be addressed alongside efforts to enhance adhesion [150,151].

The additives used to enhance the adhesive properties of hydrogels have been widely studied [152]. In terms of enhancing hydrogel adhesion, in addition to synthetic polymers as additives, natural polymeric materials such as starch have been widely used to enhance the adhesion properties of hydrogels. Qin et al. [153] fabricated a composite hydrogel patch (CPB) by introducing black phosphorus nanosheets. As shown in Figure 6a, the hydrogel patch is composed of methacrylated hyaluronic acid (HAMA), gelatin, and polyvinyl alcohol (PVA). The internal tri-crosslinked network of the hydrogel provides its mechanical properties. The doping of black phosphorus, which possesses strong water absorption capabilities, can effectively eliminate the hydration layer between tissues and adhesives, achieving excellent wet adhesion performance of the composite hydrogel. Compared to the hydrogel patches without black phosphorus nanosheets (CP), the wet tissue adhesion effect was more significant. Since black phosphorus automatically degrades into phosphate anions, such as PO_4_^3−^, when it encounters oxygen and water, these anions can interact electrostatically with groups contained within other polymers in the CPB patch, such as HAMA, gelatin, and PVA, which contain cationic groups. This interaction disrupts the self-assembled structure of the polymer chains, leading to their dissociation and swelling, thus enhancing water absorption. In Figure 6b, the shear stress of CP and CPB adhering to various organs such as the skin, heart, stomach, and liver of pig tissue is shown. It can be observed that the shear stress on wet pig skin can reach 171 kPa, and similarly, the adhesion capacity on other organs also shows excellent results. Figure 6c displays the physical photos of the shear test, clearly showing the firm adhesion of CPB patches to the tissue. Figure 6d demonstrates the interfacial toughness of CP and CPB adhered to various organ tissues (skin, heart, stomach, liver) inside the pig’s body, reaching 457 N/m between wet pig skin interfaces. Figure 6e presents the photos of the 180° peel test. The enhancement of adhesive performance by black phosphorus nanosheets is directly evidenced by the above data. Such an excellent substance for enhancing wet tissue adhesion is crucial for the fabrication of hydrogels used in wearable sensors, providing a novel approach to improve the adhesive properties of hydrogels.

In addition to high adhesion on daily occasions, maintaining reliable adhesion when facing harsh conditions is also crucial for hydrogel-based wearable sensors. Liu et al. [154] have developed a multifunctional organic hydrogel that demonstrates not only strong wet adhesion ability but also possesses an outstanding ability to adhere in extreme environments. As shown in Figure 6f, the hydrogel was prepared by adding chitosan (CS), tannic acid (TA), 2-methoxyethyl acrylate (MEA), acrylic acid (AA), and 2-acrylamido-2-methyl-1-propanesulfonic acid (AMPS) into dimethyl sulfoxide (DMSO), The role of AMPS and TA is crucial, as the strong hydrogen bonds formed between them provide structural integrity of the hydrogel in extreme environments. The organic hydrogel exhibited an excellent wet adhesion performance, mainly attributed to the introduction of electrostatic and hydrophobic interactions. The electrostatic interaction primarily arises from the sulfonic acid groups in AMPS, which are negatively charged and can form coordinate bonds with positively charged groups on the surface of the substrate, such as cations on metal surfaces, thereby enhancing adhesion. Hydrophobic interactions are derived from the hydrophobic groups in MEA, which form a hydrophobic layer at the adhesive interface, preventing the influx of water and thus enhancing adhesion. Additionally, the catechol groups in TA can penetrate the hydrated film and bind to the hydrophobic groups on the substrate surface, further improving the adhesive performance under wet conditions. The presence of tannic acid provided a basis for its adhesion performance, as its abundant functional groups could form coordination bonds with metal surfaces and abundant hydrogen bonds with hydroxyl, siloxane bonds, and amino groups on other substrate surfaces. Figure 6g shows the dry and wet adhesion strengths of the hydrogel at different TA contents. It can be seen that the wet adhesion strength can reach up to 72 kPa with an increasing TA concentration, demonstrating the important role of tannic acid in enhancing the wet adhesion strength of the hydrogel. Figure 6h shows that the hydrogel adheres well to pig skin, even after being rinsed with water flow. This ensures that the hydrogel sensors can reliably adhere to the wet tissues without slipping, providing an important basis for acquiring electrophysiological signals.

The use of certain functional groups in artificially synthesized polymers to enhance the adhesion performance of hydrogels has been widely studied. Meanwhile, there are many natural phenomena of adhesive properties found in animals and plants, which can achieve strong adhesion while also possessing excellent biocompatibility. These could bring great inspiration for developing hydrogel-based wearable sensors with safe and long-term adhesion ability. Inspired by glutinous rice, Zhou et al. [155] reported a highly adhesive organic hydrogel. As shown in Figure 6i, the glutinous rice-inspired organic hydrogel was constructed by introducing branched amylopectin (AP) into the chemically crosslinked copolymer network of acrylic acid (AA) and acrylamide (AM). A “one-pot” photo-induced gelation process was used to prepare the hydrogel in a mixed solvent of glycerol and water (G-W) containing potassium chloride (KCl). The strong adhesion performance of the organic hydrogel is attributed to the abundant hydroxyl groups in AP, which increase the number of polar groups. In the meantime, the large number of glycerol molecules weakens the hydration effect, therefore exposing more polar groups on the surface and enhancing the interaction between the hydrogel with the substrate. Figure 6j shows the variations in adhesion strength with different ratios of AA and AM; the strongest adhesion strength was obtained to be around 12.5 kPa when they were in a 1:1 ratio. The robust adhesion performance of this hydrogel has been verified in various substrates, including pig skin, glass, and many other materials. And the adhesion was also proved whether subjected to sub-zero or high-temperature conditions. These stable adhesion stabilities ensure reliable sensing performance for the hydrogel to be applied in wearable sensors. The unique properties of hydrogels enable the production of wearable sensors that exhibit high durability and resistance to damage, which is of great significance for the creation of high-sensitivity sensors used to monitor the vital signs and movements of underwater workers.

### 3.3. Mechanical Properties

Except for the necessary conductivity and adhesion properties, excellent mechanical performance is also a significant property for the hydrogels as an ideal candidate for the preparation of stretchable wearable sensors. High stretchability and toughness are among the basic mechanical characteristics required for hydrogel-based wearable sensors [156], which enable the sensors to effectively accommodate body deformations and maintain good conformal contact, thereby ensuring the stability of signal transmission. Meanwhile, a lower modulus allows the hydrogels to better conform to human skin and tissues, further enhancing wearing comfort and biocompatibility [157]. Additionally, hydrogel materials with special stress resistance, such as tolerance to torsion, shear, and puncture by sharp objects, are also of great significance for the reliability of wearable sensors [151,158]. Such hydrogels with excellent comprehensive mechanical properties not only enable sensors to better adapt to complex environments but also significantly improve their sensing stability and durability.

Excellent mechanical properties are fundamental for the fabrication of wearable sensors; they can expand the application scenarios of wearable sensors, such as implantation in the body or epidermal monitoring [135,136]. Therefore, the optimizations of fabrication methods for enhancing the mechanical properties of hydrogels have been widely studied. Gao et al. [159] utilized a solvent exchange-assisted wet annealing strategy to produce a hydrogel with robust mechanical properties. As shown in Figure 7a, the preparation process of this hydrogel begins with dissolving Poly (vinyl alcohol) (PVA) in dimethyl sulfoxide (DMSO) to form a homogeneous solution for eliminating non-covalent interactions between polymers. Subsequently, by exchanging DMSO with the solvent glycerol, the polymer solution was induced to transform into an organic gel with a crosslinked polymer network. Next, wet annealing was performed on the PVA/glycerol organic gel to adjust the conformation of the polymer chains and enhance the structural density. Finally, through a second solvent exchange, the organic gel was transformed into a hydrogel with high-chain entanglement and crystallinity, resulting in ultra-high toughness. This novel preparation method greatly enhances the mechanical properties of the hydrogel samples. Figure 7b shows the stress–strain curves of the hydrogel samples undergoing wet annealing at 135 °C for varying durations between 5 min and 120 min. It is observed that the tensile properties of the hydrogel samples progressively improved with increasing annealing time. Through repetitive experimentation, exceptional mechanical performances, including a tensile strength of 5.11 ± 0.16 MPa, a toughness of 51.66 ± 1.86 MJ m^−3^, a stiffness of 0.76 ± 0.05 MPa, and a fracture strain of 1879 ± 10%, were achieved when the PVA content was 15%. The exceptional mechanical properties of the hydrogel are visually demonstrated in Figure 7c. It can be seen that the hydrogel sample can support two dumbbells with a total weight of 12 kg, which exceeds the weight of the sample itself by 12,000 times. In contrast to traditional epidermal sensors, this hydrogel sensor, with exceptional mechanical properties, is suitable for implantation within the body. Their superior mechanical resilience ensures that the sensors remain unaffected by the acidic and alkaline fluids characteristic of the body’s internal environment, thereby preventing functional failure.

The modulus is one of the important factors when considering the mechanical properties, and preparing hydrogels with a modulus that matches the skin, tissues, and nerves of the human body is crucial for electrophysiological signal monitoring. If the modulus of the hydrgel does not match that of the biological tissue, it may lead to serious consequences such as tissue or skin damage. Therefore, it is desirable to develop hydrogels with tunable moduli for the preparation of wearable sensors. Recently, Chen et al. [160] prepared a hydrogel with a tunable modulus. As shown in Figure 7d, firstly, poly(N-isopropylacrylamide) (PNIPAM) microgels were synthesized via a precipitation copolymerization method and functionalized by introducing vinyl groups into the microgel network. Subsequently, by using potassium persulfate (KPS) and tetramethyl ethylenediamine (TMEDA) as initiators, a free-radical polymerization reaction was conducted with acrylamide (AAm), lauryl methacrylate (LMA), and the functionalized microgels to prepare a hydrogel network. The addition of microgels plays a significant role in promoting the crosslinking process of hydrogels. Specifically, an increase in the content of microgels markedly enhances the internal crosslinking density of hydrogels, thereby endowing them with higher moduli. Consequently, by finely tuning the proportion of added microgels, the crosslinking density of hydrogels can be effectively controlled, allowing for the precise modulation of their modulus among a highly tunable range (47.0 kPa–59.7 kPa) that matches the modulus of elasticity of biological neural tissue. As illustrated in Figure 7e, the stress–strain curves of hydrogel samples with different microgel contents demonstrate that the tensile properties of the hydrogel are enhanced from an initial 1410% to 1850% with the increase in microgel content. Figure 7f demonstrates the photographs of the hydrogel in multiple mechanical property tests to show the comprehensive mechanical performance. The exceptional toughness is displayed by lifting the weight plate. Its excellent puncture resistance is proved by the puncture test with a screwdriver. Its high scratch resistance is verified by the cutting test with a scalpel with no scratches left. Finally, by inflating it with nitrogen to form a balloon shape, its good stretchability is demonstrated. The exceptional mechanical properties of hydrogels greatly expand their range of applications. Their superior modulus tunability and high degree of compatibility with tissue modulus make them very suitable for the production of wearable sensors for in vivo implantation.

Nanomaterials possess diverse functionalities, and appropriate addition to hydrogel systems can significantly enhance their electrical properties and have a significant impact on their mechanical properties. Through appropriate stretching, the nanomaterials in the hydrogel system can be induced to be arranged in an orderly manner, thus significantly enhancing the mechanical properties of the system. This process effectively alleviates the stress concentration phenomenon at the crack tip, which endows the hydrogel with high sensitivity to crack extension and excellent fatigue resistance. Li et al. [161] prepared a highly conductive tough hydrogel by adding silver nanowires (AgNWs) and liquid metal (LM) as nanofillers into the hydrogel system. Figure 7g illustrates the preparation process of the hydrogel. Firstly, LM particles with an average diameter of 0.6 µm were obtained by ultrasonic dispersion in ethanol. Then, LM particles and AgNWs were added to a PVA solution at 95 °C to prepare a precursor solution for hydrogel. The precursor solution was transferred to a silicone mold to cure and form PVA-AgNWs-LM (PAL) hydrogel. Hydrogen and coordination bonding interactions between liquid metal LM and poly(vinyl alcohol) PVA molecular chains can effectively relieve the stress concentration at the crack tip of the hydrogel, which provides the hydrogel with resistance to crack extension and fatigue. As the LM content increases, these interactions and the dilution of the density of polymer chain segments lead to the enhancement of the elongation at break and strength of the hydrogel. The stress–strain curves of the hydrogel samples made with different contents of LM are presented in Figure 7h. The best stretchability was obtained at the mass ratio of LM and PVA 1:1, reaching 5237.7 ± 74.6%. Figure 7i shows the photographs of the hydrogel sample stretched by 5000%. The crack extension stress–strain curves of PVA hydrogels with different LM contents are further exhibited in Figure 7j. It can be seen that hydrogel samples with incisions can withstand a maximum fracture strain of up to 2145%, which also confirms the crack extension insensitivity as well as the great fatigue resistance of the hydrogel. Figure 7k shows the Young’s modulus of hydrogel samples with different LM contents. It can be seen that the modulus has a decreasing trend with an increase in the LM content, achieving an adjustability from 48.8 ± 1.6 MPa to 12.2 ± 0.2 MPa. Such hydrogel sensors are well-suited for monitoring physiological signals, remaining stable even during high-intensity athletic activities, thus providing accurate data support for performance analysis and injury prevention. Table 2 summarizes the different mechanical properties of stretchable hydrogel wearable sensors with various doped nanomaterials for physiological signal monitoring.

## 4. Electrophysiological Monitoring Applications

Stretchable hydrogel-based wearable sensors have demonstrated tremendous potential in the field of health monitoring, with successful development for the detection of physiological electrical signals, blood pressure, respiration, and motion tracking [162,163,164]. Their superior performance is crucial for the timely and accurate monitoring of electrocardiogram (ECG) signals, providing an effective solution for the prevention, diagnosis, and treatment of life-threatening cardiac conditions such as myocardial infarction [165]. Electromyogram (EMG) signals also reflect a wealth of information about the wearer’s immediate movements. This is of significant importance for providing professional motion guidance to athletes and preventing sports injuries [166]. Similarly, electroencephalogram (EEG) and electrooculogram (EOG) signals are vital indicators of human activity and health status. By detecting anomalies in EEG signals, we can promptly identify diseases, which is crucial for preventing casualties and ensuring public health [167,168]. In summary, hydrogel-based wearable sensors are of paramount importance in the monitoring of physiological signals for the preservation of human health and the prevention and treatment of diseases.

### 4.1. ECG Monitoring

Electrocardiogram (ECG) signals play a crucial role in the medical field, being widely used for the diagnosis of heart diseases, assessment of cardiac health, monitoring of cardiac activities, and guiding treatment strategies. However, current ECG monitoring sensors face several challenges, such as their low sensitivity, poor adhesion to tissue, and inability to be used for extended periods. Hydrogels possess unique properties, such as flexibility and conductivity, and to address these issues. Hydrogel-based wearable ECG monitoring sensors have now undergone a revolutionary transformation compared to traditional rigid sensors. They offer a higher signal-to-noise ratio and sensitivity, effectively distinguishing between ECG signals and background noise, thereby enhancing the accuracy and reliability of monitoring [169], thus providing more accurate data support for clinical diagnosis and treatment. In addition, these sensors maintain stability in harsh environments, ensuring reliable monitoring performance, even in humid or underwater conditions [170]. Lastly, the new generation of hydrogel-based ECG monitoring sensors has seen a notable improvement in biocompatibility, better adhering to human tissue and reducing discomfort during use [171]. Researchers have achieved remarkable results by developing high-performance hydrogel-based ECG signal detection sensors for both epidermal and implantable applications [172,173,174].

Hydrogel sensors offer significant performance advantages over other types of sensors in terms of ECG detection. Compared to the metallic ECG monitoring sensors, hydrogel sensors are lighter, low cost, simple to prepare, and suitable for wearable and in vivo implantation. Compared to silicon-based sensors used for blood pressure monitoring, hydrogel sensors exhibit greater flexibility and biocompatibility, adapt to differently shaped surfaces, and do not require additional coatings. Hydrogel sensors are also self-healing. Although they may malfunction in dry environments and are less conductive and stable than traditional inorganic sensors, hydrogel sensors are preferred for applications requiring high flexibility, biocompatibility, and environmental sensitivity.

Myocardial infarction (MI) is a serious heart disease that can potentially be life-threatening. Therefore, it is crucial to conduct real-time monitoring of electrocardiogram (ECG) signals for the patients and ensure timely rescue after the MI happens. Qiu et al. [175] developed an implantable curcumin-nanocomposite ionic conductive hydrogel based on a shell–layer structure for the diagnosis and treatment of myocardial infarction. Figure 8a presents the application of the prepared curcumin-nanocomposite ionic conductive hydrogel as a flexible strain sensor on the surface of rat hearts to collect cardiac signals. The hydrogel can sensitively detect small deformations caused by heartbeats, displaying stable, repeatable resistance signal waveforms. It can also differentiate between signals from normal hearts and hearts with myocardial infarction. Figure 8b shows the electrocardiogram signals of normal rats and rats with myocardial infarction, along with the relative resistance changes ΔR/R_0_ of the hydrogel bioelectronic patch. It can be observed that the signals measured by the hydrogel patch have corresponding peaks in resistance changes compared to the standard electrocardiograms, including for the P-wave, QRS-wave, and T-wave. The hydrogel sensor patch could accurately capture cardiac signals and detect abnormal signals in ischemic and infarcted myocardium models. When applied to infarct sites, the hydrogel sensor exhibited weaker or absent resistance signals due to myocardial damage. This indicates the hydrogel sensor patch’s reliable diagnosis of myocardial infarction and demonstrates its potential as an alternative to traditional ECG monitoring. The results suggest that the implantable hydrogel sensors can reliably diagnose heart diseases and potentially replace traditional ECG monitoring, especially in detecting myocardial infarction.

For the accurate monitoring of ECG signals, the effective integration of hydrogel wearable) sensors with the heart surface is a crucial issue to be addressed. Considering the heart operates in a complex physiological environment, hydrogel-based sensors need to maintain stability in humid conditions while possessing robust electrical properties. Sun et al. [41] recently developed a functionalized hydrogel patch for detecting physiological signals of cardiac mechanical deformation. By modifying the main chain of polyaniline (PAN), they grafted boronic acid and carboxylic acid side chains onto the PAN backbone, thereby endowing the polymer with multiple new functionalities, including enhanced conductivity, hydrophilicity, and self-healing performances. The entire process of its intrinsic mechanism is illustrated in Figure 8c; it can be seen that the hydrogel patch could convert the cardiac mechanical deformation into resistance waveform changes. Modifying the polyaniline backbone can effectively remove complex fluids from the surface of the heart and form an interlocking structure, thus tightly integrating the hydrogel patch with the myocardial surface. The sensing platform based on the hydrogel can continuously record cardiac mechanical physiology and monitor the amplitude and rhythm of abnormal hearts with myocardial infarction. Figure 8d exhibits the representative electrocardiograms (ECGs) of rats at 3, 14, and 28 days post-surgery for the myocardial infarction group, the PVA hydrogel control group, and the patch hydrogel group, all showing a significant elevation of the ST segment, indicating acute myocardial injury. After myocardial infarction, fibrosis in cardiomyocytes significantly impedes the generation and propagation of action potentials, potentially resulting in severe cardiac pathologies and endangering life. Promptly and effectively addressing the electrical conduction deficits caused by post-infarction cardiomyocyte damage is vital for health preservation. The spread of calcium ions serves as an indicator of cardiomyocyte electrical conductivity. Figure 8e displays a heatmap of the activation times as calcium transient signals extend from the cardiac apex to various heart regions. The figure indicates that the hydrogel patch enhances electrical conduction and effectively mitigates the electrical conduction impairment in the infarcted zone. Calcium ion conduction in areas covered by the patch is notably superior to that in regions compromised by myocardial infarction. This finding confirms the hydrogel patch’s superior capability in restoring electrical conduction, which is essential for the repair of infarcted myocardium. Figure 8f displays the current curves at 3, 14, and 28 days after myocardial infarction, reflecting the strain of the hydrogel patch during heart contraction and relaxation. The hydrogel-based ECG monitoring sensor has exhibited exceptional durability with a month of uninterrupted and consistent monitoring, affirming its efficacy for prolonged physiological electrical monitoring within an implantable in vivo setting. This innovative technology has the potential to deliver continuous and precise health data to enhance diagnostic precision and treatment planning, thereby playing a pivotal role in the prevention, diagnosis, and monitoring of various cardiovascular conditions.

Post-myocardial infarction cardiac repair is an urgent challenge in medicine. Microcirculation damage caused by myocardial infarction leads to local tissue necrosis and fibrosis, which may trigger malignant arrhythmias and changes in the myocardial structure. If not promptly addressed, the impairment of myocardial cells’ normal functions may severely impact human health. Consequently, the issue of repairing the heart following a myocardial infarction is an urgent and critical challenge that needs to be addressed. To address this challenge, Xing et al. [176] proposed an innovative microchannel hydrogel suture thread aimed at promoting cardiac recovery after myocardial infarction through bidirectional signal transmission, on-demand drug release, and cardiac monitoring. The preparation of hydrogel sutures involves dissolving polyvinyl alcohol (PVA) in water to form a solution, which is then uniformly coated onto nylon fiber molds using a stepper motor. To impart conductivity, a polypyrrole (PPy) solution is mixed with the PVA solution, and the coating process is continued. After drying the coated molds, the hydrogel is formed through a treatment with sodium hydroxide, following which the molds are peeled off to obtain the hydrogel sutures. Figure 8g shows the prepared hydrogel suture thread used for monitoring different tissues and demonstrates the process of transmitting myocardial infarction signals. This hydrogel suture thread is not only suitable for various tissues such as muscles, skin, heart, colon, and blood vessels but can also transmit physiological signals generated by the heart wirelessly via Bluetooth to digital devices such as mobile phones, enabling the real-time monitoring of heart health, as shown in Figure 8h. Furthermore, Figure 8i illustrates the comparison between the electrocardiogram signals recorded by this device and those obtained by commercial medical devices after one month of measurement in each group of rats. The similar trends between the two sets of signals indicate the excellent performance of these devices in monitoring electrocardiogram signals. This innovative technology holds promise as an effective tool for treating myocardial infarction and provides a new pathway for cardiac rehabilitation.

### 4.2. EMG Monitoring

Muscles play vital roles in various parts of the human body, making the development of corresponding electromyography (EMG) monitoring crucial for fields such as rehabilitation therapy, biomechanical research in sports, intelligent prosthetics, and athletics. The hydrogel-based wearable sensors for EMG signals monitoring are expected to possess outstanding durability, stretchability, breathability, and excellent biocompatibility [5,177,178]. Additionally, their excellent electrical properties [169,179] will contribute to accurately monitoring the deformation of different parts of the body muscle, meeting various monitoring demands. The advancement of hydrogel-based EMG sensors is expected to provide more precise data support for human movement and rehabilitation.

The non-smooth irregular surfaces of the human epidermis pose challenges for the design of hydrogel-based electromyography (EMG) sensors, as it requires sensors to make sufficient contact with the biological surface and maintain their shape to ensure accurate signal acquisition under complex stress conditions. Simultaneously, considering biocompatibility with the skin is also critical to prevent potential skin allergies or inflammatory reactions. Recently, Xia et al. [37] proposed an in situ-formed hydrogel electrode (ISF-HEs). The ISF-HEs are fabricated by dissolving, mixing, and self-polymerizing materials, including glycidyl methacrylate (GMA), methacrylic acid (MA), dodecyltrimethylammonium chloride (DA), and carbon nanotubes (CNTs). It can maintain high shape conformability with the contact surface at skin bends and wrinkles, achieving synchronous deformation with the skin, and possesses an extremely short response time and excellent biocompatibility. Figure 9a demonstrates a comprehensive shooting assessment system based on the hydrogel sensor, which can comprehensively display the activities of various muscles in the right arm during rifle shooting. In Figure 9b, synchronous surface electromyography signals (EMG), obtained using the ISF-HEs and commercial Ag/AgCl electrodes, are compared. The results indicate that compared to commercial electrodes, ISF-HEs have fewer losses of electromyography signals and superior stability. Additionally, Figure 9c shows the EMG signals of the wrist flexor muscle during the preparation and firing processes, as well as the myoelectricity signals caused by wrist joint flexion. These signal waveforms clearly distinguish between the different actions of the arm, further confirming the potential prospects of the hydrogel in body motion sensing. Overall, this study provides strong support for addressing the issues of interference caused by motion artifacts in EMG signal monitoring, and its findings are expected to contribute to progress and innovation in fields such as medical care, rehabilitation therapy, and biomechanical research in sports.

The development of wearable sensors is usually constrained by various factors such as manufacturing processes, materials, and biocompatibility. The complex structure of human skin contains various biological sensors, which can react correspondingly to different changes in surrounding environments, playing a significant role in human perception, motion reaction, and object recognition. Inspired by the human skin, Han et al. [38] developed a hydrogel-based wearable skin sensor that achieves the perception of various biophysical signals by imitating the functions, structures, and material properties of human skin receptors. The Polyaniline-Poly(vinyl chloride) (PANi-PVC) ionic hydrogel mimics the slow adapting (SA) receptors’ response to static pressure, while the Poly(vinylidene fluoride-trifluoroethylene) (PVDF-TrFe) hydrogel simulates the rapid adapting (RA) receptors’ reaction to dynamic pressure. The sensor surface is designed with a conical structure to increase the contact area with the skin and enhance sensitivity. The PANi-PVC ionic hydrogel possesses conductivity, enabling the measurement of ECG signals. Furthermore, the PVDF-TrFe hydrogel exhibits piezoelectric properties, converting mechanical stimuli into electrical signals. Specifically, this sensor, when worn on the wrist, can monitor four functions, including electromyogram, blood pressure, electrocardiogram, and neural signals, enabling comprehensive detection and the tracking of health conditions. As shown in Figure 9d, the wearable sensors were immobilized on the calf and wrist to record various electromyographic signals accurately. Figure 9e, respectively, shows the corresponding electromyogram (EMG) signals obtained using the prepared hydrogel sensor when the subject is standing, sitting, and kicking. It is clear that the signals corresponding to different motions of the subject exhibit distinct peaks, and there are significant differences in the peak values between different actions. And the electromyography signals of the wrist while continuously increasing gripping force are shown in Figure 9f. It can be observed that as the gripping force gradually increases, the peak value of the signal also increases. In summary, this hydrogel wearable sensor can not only accurately distinguish different actions of the human body in real-time but can also differentiate between different magnitudes of the same action, playing an important role in monitoring human health conditions during sports.

For hydrogel-based wearable sensors attached to the skin surface to acquire EMG signals, it is essential not only to consider the signal quality and monitoring accuracy but also to pay attention to the biocompatibility issues caused by their contact with the human body surface. Ensuring that the hydrogel sensor is harmless to human health is the most basic and necessary requirement. Recently, Zhang et al. [180] reported the synthesis of a conductive hydrogel by mixing Poly(3,4-ethylenedioxythiophene)-Poly(styrenesulfonate) (PEDOT:PSS) solution, zwitterionic [2-(methoxyethoxy)ethyl]dimethyl-(3-sulfopropyl) ammonium (SBMA), and poly(ethylene glycol) diacrylate (PEGDA) in appropriate proportions, followed by a spontaneous polymerization process at room temperature. The obtained hydrogel possesses excellent biocompatibility, outstanding antibacterial properties, and the ability to monitor physiological signals effectively. Its outstanding antibacterial properties stem from the addition of poly[2-(methacryloyloxy)ethyl]dimethyl-(3-sulfopropyl) (PSBMA), which can disrupt the cell membranes of bacteria through charge interactions, thereby killing bacteria. The experimental results revealed that the hydrogel exhibited a high bactericidal efficacy of 97% against Staphylococcus aureus and significant antibacterial activity against Escherichia coli. Figure 9g shows the monitoring of electromyography signals with the hydrogel sensor attached to the densely hairy calf when volunteers squat and stand up. The electromyography signals obtained by the hydrogel electrode and commercial electrodes, which were attached to the calf after several squatting cycles, are shown in Figure 9h. It can be seen that the commercial electrode begins to exhibit signal noise after several cycles and completely fails after 38 cycles, while the signals from the hydrogel sensor remain stable throughout the testing process, demonstrating excellent stability of the prepared hydrogel sensor patch. The exceptional antibacterial properties, high biocompatibility, and signal stability exhibited by this hydrogel confer tremendous potential and promising prospects in key areas such as sports health monitoring, medical rehabilitation, and exercise tracking.

### 4.3. EEG and EOG Monitoring

Electroencephalogram (EEG) technology is a method that records and monitors brain activity through electrophysiological indicators. When the brain is active, the electrophysiological activities of brain neurons produce detectable signals on the cerebral cortex and scalp surface. By precisely recording these EEG signals, brain activities can be decoded, which is crucial for patients with limited expressive abilities due to diseases. Timely and effective monitoring of electroencephalogram (EEG) signals can also prevent and diagnose serious threats to health, such as epilepsy, Parkinson’s disease, and stroke. Hydrogels are widely used in EEG detection due to their flexibility, stretchability, and excellent electrical properties. Smart hydrogel sensors have been developed that can acquire high-quality EEG signal images and continuously and stably collect signals for over ten hours [181,182]. Electrooculography (EOG) is a technique that detects eye movement and potential changes by recording the electrical signals generated by ocular movements. It is primarily utilized in the study of eye movement control and visual attention. By accurately monitoring EOG signals, doctors can gain a deeper understanding of the complex functions of the visual system and effectively diagnose and treat a range of related diseases, including sleep disorders, oculomotor disorders, and neurodegenerative diseases. At the same time, the proper monitoring of electrooculogram (EOG) signals also holds profound significance and plays a crucial role in human–machine interfaces, virtual reality, wireless remote control, etc. [183].

Natural material-based hydrogels have garnered significant attention from researchers in recent years due to their excellent biocompatibility and environmental sustainability. Among them, some natural factors with special functionalities produced by living organisms have been discovered and applied in the synthesis of hydrogels. Inspired by the natural moisturizing factor (NMF) in the skin, Ying et al. [40]. developed an all-natural ionic biological hydrogel mainly composed of gelatin and pyrrolidone carboxylic acid sodium salt (PCA-Na). This hydrogel exhibited outstanding dehydration resistance and biodegradability and was proven to be a reliable biological interface. Consequently, this hydrogel was used to fabricate a wearable electrophysiological sensor for long-term EEG signal monitoring. As shown in Figure 10a, an integrated wearable EEG acquisition headband was prepared using the hydrogel. The hydrogel electrode directly contacted the skin to facilitate the extraction of EEG signals. In Figure 10b, the EEG signals monitored using the prepared hydrogel electrodes and commercial EEG electrodes are displayed. The figure illustrates that the alpha rhythm of the EEG waves monitored by the hydrogel electrodes exhibited higher fidelity than those measured by commercial electrodes. Moreover, after the test, the hydrogel electrodes could be easily removed with warm water without leaving any residue or causing skin allergies, demonstrating the reliability of the hydrogel electrodes and the excellent biocompatibility provided by natural biological materials. Figure 10c compares the blink artifacts in the EEG signals measured with and without the hydrogel electrodes. The comparison clearly shows that the electrodes with hydrogel significantly reduced blink artifacts in the EEG signals, thereby improving the quality and reliability of the monitored EEG signals. The superior biocompatibility offered by natural biological materials allows hydrogel sensors to better adapt to applications on the body surface and within the body, promising a wide range of applications in medical monitoring, disease treatment, and non-contact sensing.

Long-term, persistent, and reliable physiological signal monitoring has always been a challenging problem to solve. However, due to the natural tendency of hydrogels to lose water, preparing wearable hydrogel sensors for persistent electrophysiological signal monitoring poses certain problems. To address this long-term electrophysiological monitoring issue and improve the user experience, Wang et al. [39] designed and developed an ultra-thin (approximately 10 μm), breathable hydrogel film that can be used for continuous electrooculogram (EOG) monitoring for more than 8 days. The hydrogel film was prepared by crosslinking polyurethane (PU) nanonetwork and gelatin by utilizing the electrospinning technique. The hydrogel features an ultra-thin profile that can attach seamlessly to the skin while also allowing for adequate air permeability, thereby ensuring wearer comfort. Utilizing an enhancement layer crafted through electrospinning techniques, the hydrogels exhibit exceptional mechanical robustness and resilience, which guarantees the longevity of the sensor during continuous usage. To preserve the hydrogel’s hydration, substances like glycerol are incorporated into the system, preventing signal degradation. Figure 10d illustrates the wearing diagram of the hydrogel sensor during an electrooculogram (EOG) signal recording. The hydrogel sensor can be directly worn around the eyes without complex additional steps. The EOG signals collected using the developed ultra-thin hydrogel sensor and commercial gel sensor are, respectively, shown in Figure 10e. Through a continuous 24-h monitoring period, it was observed that the signals recorded with the ultra-thin hydrogel electrodes closely mirrored those captured by commercial electrodes, displaying a high degree of alignment in their characteristic peaks in the initial period. Nevertheless, the signal quality from the commercial electrodes progressively deteriorated with the elongation of the monitoring period. In comparison, the ultra-thin hydrogel electrodes proved to be superior in terms of long-term signal stability and fidelity. Figure 10f presents the visually evoked potential (VEP) signals recorded using the ultra-thin hydrogel film. Visual evoked potentials (VEPs) represent the electrophysiological signals that track neural transmission from the retina to the visual cortex. Among the VEP signal components, the P100 wave peak is a distinctive feature that signifies the duration of visual information transmission from the retina to the brain. The developed hydrogel exhibited a consistent recording of the P100 waveform following 24 h of uninterrupted wear, thereby preserving exceptional signal stability. This study demonstrates the versatility of hydrogel sensors that are capable of long-term, high-quality monitoring of various electrophysiological signals.

When preparing hydrogel-based sensors for human physiological electrical monitoring, a challenging problem often encountered is that the hair on the human skin surface impedes proper adhesion of the hydrogel-based sensor to the skin, which could cause several serious consequences. Firstly, the monitored physiological signals may be severely distorted, making it difficult to obtain high-fidelity signals. Secondly, hair impedes the adhesion effect of the sensor to the skin surface, making it easy for the sensor to peel off and unable to perform long-term stable monitoring. To address these issues, Someya et al. [184] developed a conductive biological gel that can be applied to the skin, effectively solving the interface problem when performing physiological electrical monitoring on skin surfaces with dense hair. The conductive hydrogel was made from gelatin, sodium chloride, sodium citrate, and glycerol. At high temperatures, the hydrogel exhibits a liquid state, enabling it to readily permeate through hair and achieve seamless conformal contact with the hairy skin. Subsequently, upon contact with the skin surface, the liquid hydrogel rapidly undergoes in situ gelation, forming a connection with robust mechanical strength. This process effectively avoids the disruption caused by hair during signal acquisition, thereby ensuring the accuracy and stability of EEG monitoring. It was found that this hydrogel can achieve continuous, high-quality EEG recordings within a few days and has a very reliable biocompatibility without causing allergic reactions or inflammation in the skin tissue. As shown in Figure 10g, the upper figure shows the EEG signals recorded using the biological gel sensor during both eyes open and closed conditions, and the lower figure shows their corresponding spectrogram. It can be clearly seen that the clear alpha waves during closed eyes indicate that the biological gel sensor can capture high-resolution physiological signals, even accurately capturing very subtle movements such as opening and closing the eyes. This provides endless possibilities for the fabrication of high-precision wearable physiological signal detection devices. Figure 10h shows the EEG signals recorded by the brainwave monitoring sensors coated with biological gel and commercial brainwave monitoring sensors after continuous recording for 5 h. It can be seen that the signal quality obtained by the commercial brainwave monitoring sensor is more affected by noise, while the signal quality obtained by the biological gel sensor is more stable. The preparation of this phase-change biological gel has marked a significant resolution to the hairy skin interface problem, thereby promoting the progress of hydrogel-based wearable sensor technology.

In addition to the applications in electrophysiological signal monitoring, hydrogels have found extensive potential in diverse fields. For instance, they are utilized in the fabrication of chemical sensors for environmental monitoring, detecting hazardous substances in air, soil, or water. They could also function as moisturizers in skincare and cosmetic products, offering effective skin hydration. As drug carriers, hydrogels could facilitate controlled and targeted drug release, enhancing therapeutic outcomes. With the ongoing advancement of materials science and nanotechnology, the applications of hydrogels continue to expand, bringing innovative breakthroughs to various fields.

## 5. Conclusions and Future Prospects

This paper reviews the current research progress of hydrogel-based wearable sensors doped with nanomaterials in the monitoring of electrophysiological signals. By doping with nanomaterials, hydrogels have achieved various excellent performance. Wearable physiological signal monitoring sensors, prepared using nanomaterial-enhanced hydrogels, have many advantages compared to traditional devices. These sensors can not only fit perfectly with human skin without damage, ensuring the high-quality and stable monitoring of electrophysiological signals, but they can also be used as portable devices in various scenarios and complex conditions. This paper first introduces the types of nanomaterials used to prepare hydrogels, including nanocarbon materials, nanometal materials, and two-dimensional transition metal compounds. Then, the performance of stretchable hydrogel wearable sensors and the mechanisms for their excellent performance are summarized. Afterwards, the specific applications of nanomaterial-based hydrogel-based wearable sensors in electrophysiological signal monitoring are reviewed. This review demonstrates the tremendous potential of nanomaterial-based hydrogel sensors to expand the application scenario of wearable sensors.

Although nanomaterial-based hydrogel sensors have made encouraging progress, there are still challenges and future research issues that need to be addressed. The foremost challenge is to further enhance the performance and stability of hydrogels after the addition of nanomaterials. Key to performance improvement is enhancing the hydrogel’s adaptability to complex environments, especially for implantable devices. The complexity of bodily fluids presents challenges that may lead to the structural breakdown of hydrogels and sensor failure in a short period. One important way to enhance stability is to optimize the hydrogel’s water retention properties, as good water retention is crucial for maintaining the sensor’s conductivity and mechanical performance. Additionally, the signal transmission capability may be limited by their electrical conductivity, leading to signal attenuation or distortion. Integration into existing medical devices may also encounter compatibility issues. In future research, the broader applications of nanomaterial-based hydrogel sensors in wearable electronic systems can be further explored. For example, they can play important roles in non-contact control, virtual reality, biomimetic structures, and medical monitoring. In conclusion, nanomaterial-based hydrogels have tremendous potential in preparing wearable sensors. Through continuous exploration and research, we hope to achieve higher performance and more convenient electrophysiological signal monitoring, opening up new prospects for the development of wearable sensor technology.

## Figures and Tables

**Figure 1 nanomaterials-14-01398-f001:**
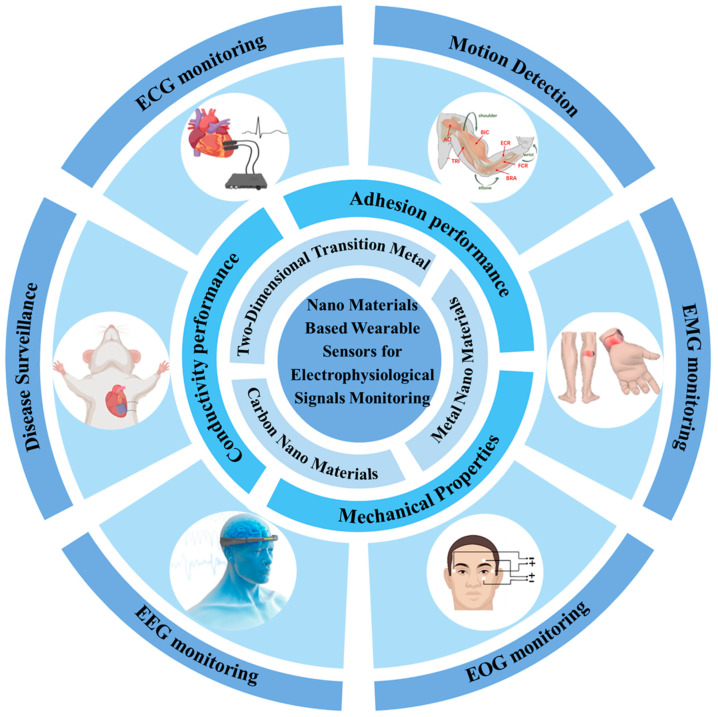
Nanomaterial-based stretchable hydrogel wearable sensor for physiological signal monitoring. Image for ‘ECG monitoring’: Reproduced with permission. Ref. [36] Copyright 2023, Springer Nature. Image for ’Motion monitoring’: Reproduced with permission. Ref. [37] Copyright 2022, American Chemical Society. Image for ‘EMG monitoring’: Reproduced with permission. Ref. [38] Copyright 2022, Wiley-Blackwell. Image for ‘EOG monitoring’: Reproduced with permission. Ref. [39] Copyright 2024, American Association for the Advancement of Science. Image for ‘EEG Monitor’: Reproduced with permission. Ref. [40] Copyright 2024, Wiley-Blackwel. Image for ‘Disease Surveillance’: Reproduced with permission. Ref. [41] Copyright 2023, Springer Nature.

**Figure 6 nanomaterials-14-01398-f006:**
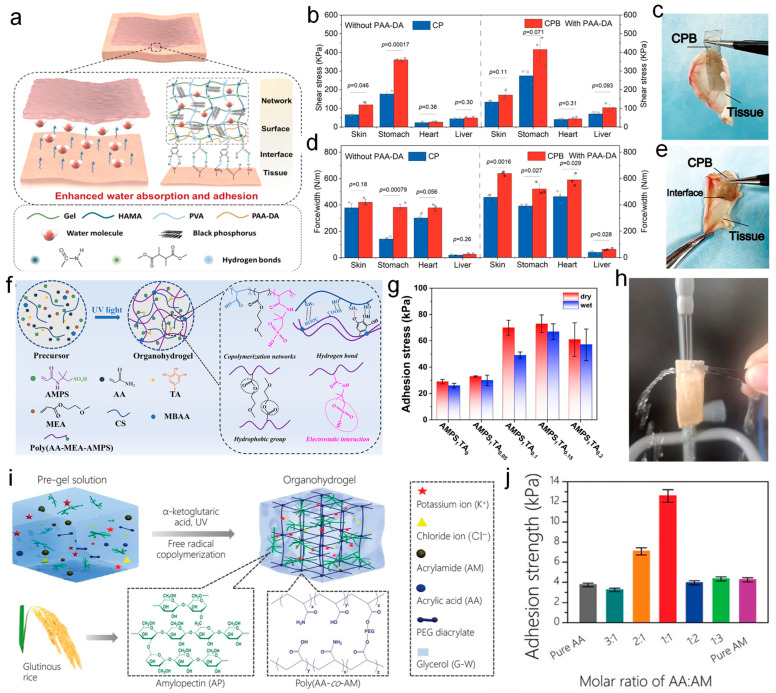
Adhesion properties of stretchable hydrogel wearable sensors. (**a**–**e**) Removal of interfacial water by adding black scale nanosheets, thereby enhancing the adhesion strength of hydrogel-based wearable sensors to various tissues [153]. (**f**–**h**) Enhancement of substance-to-substance interactions by introducing TA significantly improves the adhesion strength of hydrogel-based wearable sensors in harsh environments [154]. (**i**,**j**) Enhancement of adhesion strength of organic hydrogel-based wearable sensors by natural material starch [155].

**Figure 7 nanomaterials-14-01398-f007:**
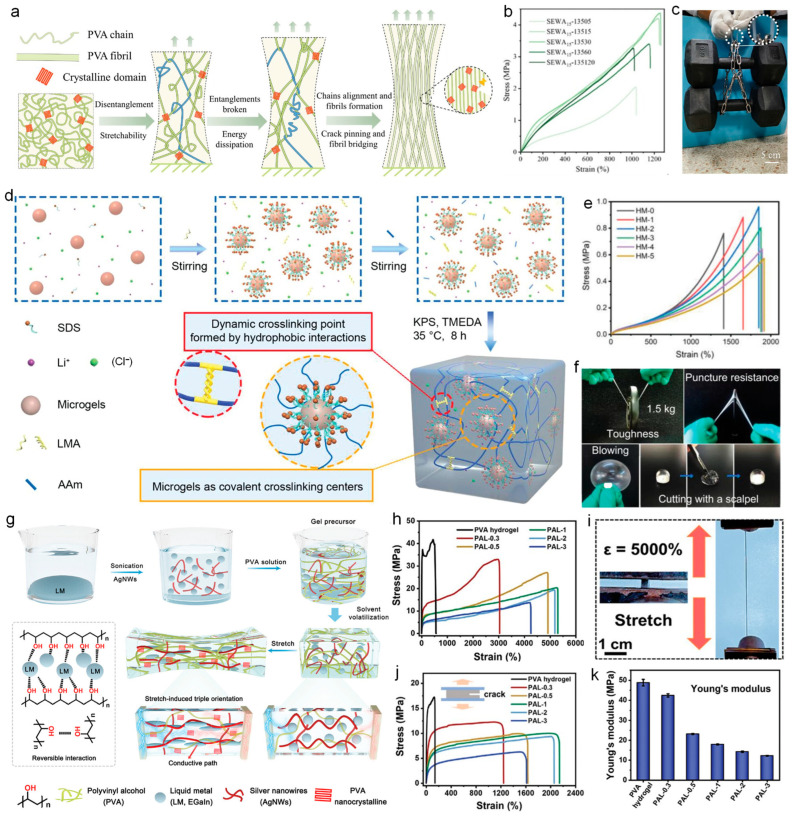
Mechanical properties of stretchable hydrogel wearable sensors. (**a**–**c**) Dramatic enhancement of the tensile strength of hydrogel-based wearable sensors by exchange-assisted wet annealing strategy [159]. (**d**–**f**) Tunable modulus of hydrogel-based wearable sensors via microgel crosslinking strategy [160]. (**g**–**k**) Significantly improving the fatigue resistance of hydrogel-based wearable sensors by stretching-induced nanomaterial ordering [161].

**Figure 8 nanomaterials-14-01398-f008:**
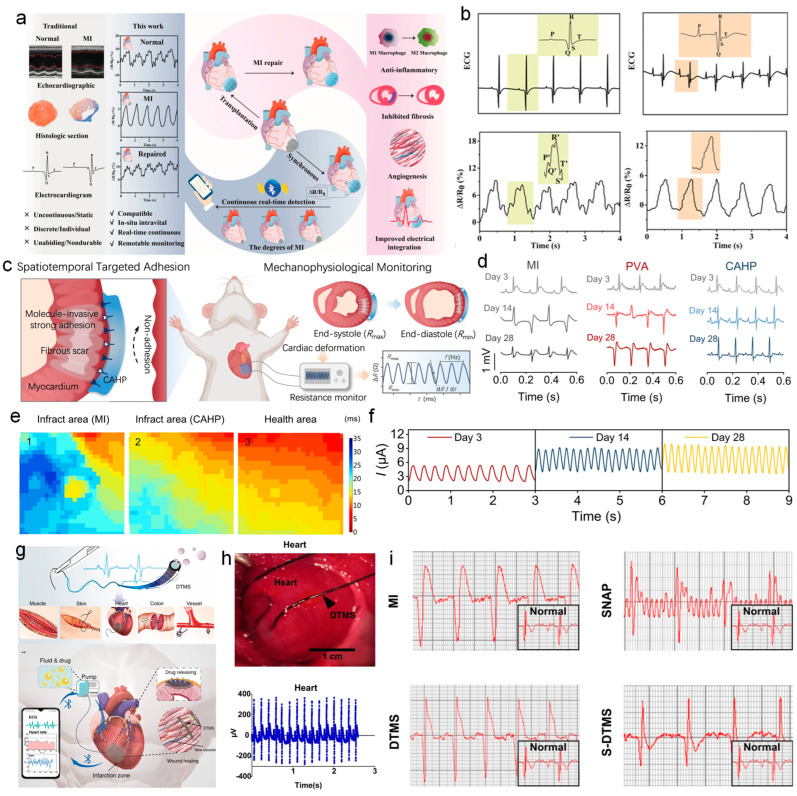
Stretchable hydrogel wearable sensor for ECG signal monitoring (**a**,**b**) Implantable curcumin-nanocomposit--ionized hydrogel patch for myocardial infarction diagnosis [175]. (**c**–**f**) Functionalized hydrogel patch with good durability and high sensitivity under complex physiological environment for ECG signal detection [41]. (**g**–**i**) Microchannel hydrogel sutures for repairing damaged cardiomyocytes after having suffered from heart-like diseases [176].

**Figure 9 nanomaterials-14-01398-f009:**
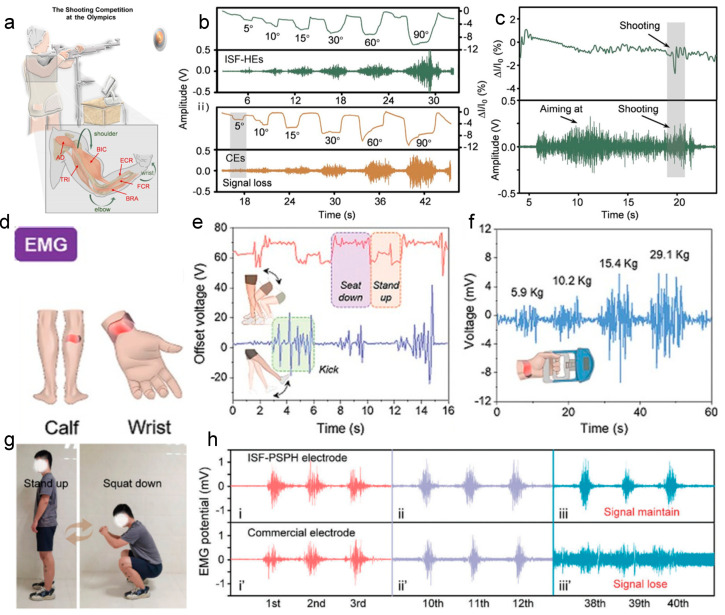
Stretchable hydrogel wearable sensors for EMG signal monitoring: (**a**–**c**) in situ-formed hydrogel electrodes with high skin fit and good biocompatibility for motion tracking [37]. (**d**–**f**) Skin-inspired ionic hydrogel wearable sensors for human motion recognition [38]. (**g**,**h**) Hydrogel patch sensors with excellent antimicrobial properties and long-term durability for health monitoring, the Roman numerals represent the contrast between the two detected signals at different times [180].

**Figure 10 nanomaterials-14-01398-f010:**
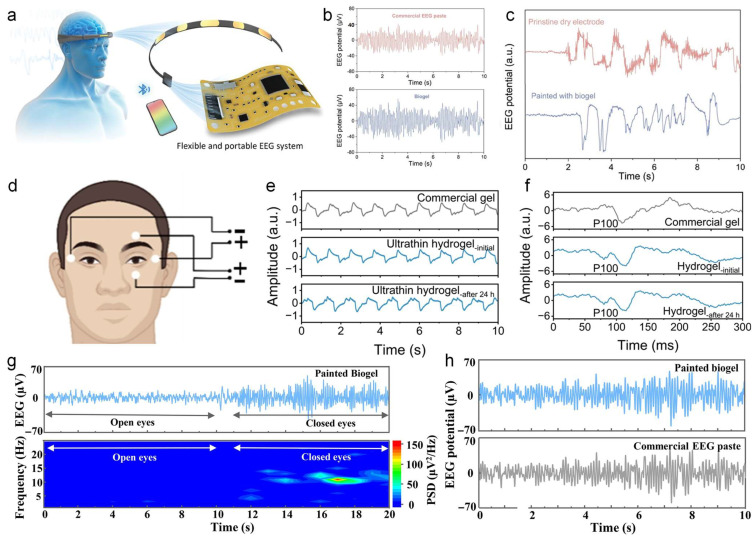
Stretchable hydrogel wearable sensors for EEG and EOG signal monitoring. (**a**–**c**) Natural material-based and long-term moisturizing hydrogel for EEG signal monitoring [40]. (**d**–**f**) Ultra-thin, breathable hydrogel film for EOG signal monitoring [39]. (**g**,**h**) Conductive biohydrogel for EEG signal monitoring in hair-intensive areas such as the scalp [184].

**Table 1 nanomaterials-14-01398-t001:** Properties of nanomaterial-based stretchable hydrogel sensors for electrophysiological monitoring.

Filled Nanomaterial	Tensile Strain %	Breaking Strength	Electrical Conductivity	GF	Adhesion	Reference
Ag/TA@GO	1250	/	0.15 S/m	3.1	Y	[47]
Graphene Oxide nanosheets	1458	2.5 MPa	4.3 S/m	3.04	Y	[54]
CNTs	217	12.58 MPa	0.071 S/m	25.98	N	[55]
mCNT-OH	1600	556 kPa	0.0031 S/cm	6.39	N	[56]
MXene	400	0.93 MPa	8.1 S/m	1.12	N	[57]
MXene	730	0.54 MPa	0.069 S/m	4.42	Y	[58]
AgNWs	480	240 kPa	1739 S/cm	/	Y	[59]
AgFs, AgNWs	1000	5.42 MPa	83.836 S/m	87	Y	[60]
Fe_3_O_4_ nanoparticles	352	56 kPa	/	4.21	Y	[61]
Silver nanoparticles	732.9	1267.6 kPa	0.39 S/m	6.8	Y	[62]
LM/HA	2700	/	116 S/m	4.8	Y	[63]

**Table 2 nanomaterials-14-01398-t002:** Mechanical properties of nanomaterial-based stretchable hydrogel sensors for electrophysiological monitoring.

Filled Nanomaterial	Young’s Modulus	Elongation at Break	Toughness	Breaking Strength	Reference
fCNTs	10–100 kPa	1000%	400–873 J/m^3^	121 kPa	[94]
LM	49–98 kPa	2000%	1.8 MJ/m^3^	70 kPa	[107]
MXene	1.41 MPa	400%	1.98 MJ/m^3^	0.93 MPa	[57]
Graphene	/	2500%	3.11 MJ/m^3^	0.27 MPa	[77]
AgNPs	7.65 MPa	126.92%	/	20.7 ± 1.8 MPa	[112]
AgNWs	19.8 kPa	1800%	/	0.35 MPa	[32]
CNTs/SiO_2_	/	3948.37%	28.48 MJ/m^3^	1939.36 kPa	[96]
EGaIn/CNCs	/	346.35%	/	0.11 MPa	[110]

## Data Availability

No new data were created or analyzed in this study. Data sharing is not applicable to this article.

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
