# Peer review of "Recent Advances of Stretchable Nanomaterial-Based Hydrogels for Wearable Sensors and Electrophysiological Signals Monitoring"

_nanomaterials, 2024, doi:10.3390/nano14171398_

Round 1

Reviewer 1 Report

Comments and Suggestions for Authors

This thorough and detailed review of nanomaterial-based hydrogels for wearable sensors is well-sourced and written. The literature review is quite thorough and the conclusions and future research to still be addressed are appropriate. There are few instances in which the text could be further scrutinized and edited before publication. For example:

The entire phrase "nanomaterial-based hydrogels for wearable sensors" is used throughout in many sentences and could likely be condensed inmany places as to not sounds so repetitive.

Page 2, Line 81: This sentence is awkwardly phrased. Rephrasing as "Hydrogels are flexible materials..." would improve the flow of the sentence.

Page 3, paragraph starting on Line 128: Graphene does not need to be capitalized throughout.

Page 17, Line 849: should read "wearable sensors attached to the skin"

Author Response

Comment 1: The entire phrase "nanomaterial-based hydrogels for wearable sensors" is used throughout in many sentences and could likely be condensed inmany places as to not sounds so repetitive.

RESPONSE: We thank the reviewers for their valuable comments. In some places in the revised manuscript, we condensed the entire phrase “nanomaterial-based hydrogels for wearable sensors” to “nanomaterial-based hydrogels”. The changes we made are highlighted in red.

Comment 2: Page 2, Line 81: This sentence is awkwardly phrased. Rephrasing as "Hydrogels are flexible materials..." would improve the flow of the sentence.

RESPONSE: We thank the reviewers for their important comments. We have rewritten the sentence as “Hydrogels are flexible materials ......” in the revised manuscript. The changes we made are highlighted in red.

Comment 3: Page 3, paragraph starting on Line 128: Graphene does not need to be capitalized throughout.

RESPONSE: We thank the reviewers for their comments. We have changed the capitalized “graphene” to “graphene” in the revised manuscript.

Comment 4: Page 17, Line 849: should read "wearable sensors attached to the skin"

RESPONSE: We thank the reviewers for their comments. In the revised version, we have changed the original text to “wearable sensors attached to the skin”.

Reviewer 2 Report

Comments and Suggestions for Authors

Duan and co-authors provided a detailed review of hydrogels for wearable sensors and electrophysiological signals monitoring. The work is vital and provides a good view of available production techniques and utilization of environmentally friendly materials in addressing the challenge of sustainable and socially responsible hydrogels. In this respect, the review is also very timely. In my opinion, the review is of high international standard and demonstrates an extensive piece of work. From my point of view, this review can be published after addressing the minor comments as per below.

Providing a new figure demonstrating different strategies for assessing the mechanical properties would be worthwhile.

Developments on hydrogels prepared with ligand-metal chelation such as metallogels have not been covered.

Providing the pros and cons of each technique for preparing gels for such applications as a new figure would provide an easy view for the readers.

Comments on the Quality of English Language

Minor editing of the English language is required.

Author Response

Comments 1: Providing a new figure demonstrating different strategies for assessing the mechanical properties would be worthwhile.

Response: We thank the reviewer’s important comment. We have added a table in section 3.3 of the revised manuscript summarizing the mechanical properties of different hydrogels made with different nanomaterials, in order to better assess the mechanical performance of various nanomaterials-based hydrogels.

Comments 2: Developments on hydrogels prepared with ligand-metal chelation such as metallogels have not been covered.

Response: We thank the reviewer’s comment. We have added a paragraph on page 3 of the revised manuscript introducing the developments of hydrogels prepared with ligand-metal chelation and cited the related references to support the points.

Comments 3: Providing the pros and cons of each technique for preparing gels for such applications as a new figure would provide an easy view for the readers.

Response: We thank the reviewer’s comment. We have added a paragraph on page 8, end of section 2 in the revised manuscript to describe the pros and cons of each technique and various nanofillers for preparing gels used in wearable sensors.

Reviewer 3 Report

Comments and Suggestions for Authors

This is a review article providing a survery on  nanomaterials-based hydrogels for wearable sensors and electrophysiological signals monitoring. The importance of nano objects in determining the properties of these materials is stressed. The subject appears to be suitable for "Nanomaterials". The contents are well presented. The references covers very recent advances in the field. Hence, in my opinion the paper can be published in the present form.

Author Response

Response: We thank the reviewer’s comment and recognition of our manuscript, we hope the revised manuscript will meet the requirements of the reviewer and the journal.

Reviewer 4 Report

Comments and Suggestions for Authors

The authors reviewed the use of nanomaterials-doped hydrogels to manufacture portable sensors for the detection of electrophysiological signals. The challenges are hydrogel performance and stability. One of the strategies for resolution is to optimize water absorption capacity. In general, the document is well done, but it needs some changes, for example:

1.    The text mentions the application of hydrogels in portable electronic devices. How does water absorption affect the performance of these devices?

2.    Line 98 mentions the self-repair capacity of hydrogel. Explain in more detail what happens at the molecular level, as well as the benefits it may have on the performance of the material.

3.    Lines 120-121 mention that carbon materials may have physical interactions with the hydrogel. Describe what type of interactions and how they relate to the structure of the nanomaterial.

4.    In line 172, it is suggested that the type of hydrogel used for the device be mentioned. 

5.    In section 4.1, it is suggested that the comparative performance of a hydrogel sensor and other sensors be included to detect cardiovascular diseases. 

6.    It is suggested to mention if there are other potential application areas for hydrogels. 

7.    It is recommended to mention the challenges that may have the implementation of hydrogels in various medical applications.

Author Response

Comments 1: The text mentions the application of hydrogels in portable electronic devices. How does water absorption affect the performance of these devices?

Response: We thank the reviewer’s comment. We have added descriptions explaining the influence of water absorption on the performance of hydrogel wearable sensors on page 9 of the revised manuscript.

Comments 2:  Line 98 mentions the self-repair capacity of hydrogel. Explain in more detail what happens at the molecular level, as well as the benefits it may have on the performance of the material.

Response: We thank the reviewer’s comment. We have explained the details of the self-repair capacity of hydrogels at the molecular level. And the benefits it endows to the performance of hydrogels are also introduced on page 3 of the revised manuscript.

Comments 3:  Lines 120-121 mention that carbon materials may have physical interactions with the hydrogel. Describe what type of interactions and how they relate to the structure of the nanomaterial.

Response: We thank the reviewer’s comment. We have explained the detailed interaction mechanisms between the carbon materials and hydrogels, and the related structural influence of the interactions is also described on page 4 of the revised manuscript.

Comments 4: In line 172, it is suggested that the type of hydrogel used for the device be mentioned.

Response: We thank the reviewer’s comment. We have described the type of specific hydrogel used for the sensor device for readers to better understand the composition details of the hydrogel sensor.

Comments 5: In section 4.1, it is suggested that the comparative performance of a hydrogel sensor and other sensors be included to detect cardiovascular diseases.

Response: We thank the reviewer’s comment. We have added a paragraph on page 15 describing the comparative performance of a hydrogel sensor and other sensors to detect cardiovascular diseases in section 4.1 of the revised manuscript, for better support the advantages of hydrogel wearable sensors for detecting cardiovascular diseases.

Comments 6: It is suggested to mention if there are other potential application areas for hydrogels.

Response: We thank the reviewer’s comment. We have added the descriptions to introduce other potential application areas for hydrogels on page 21 of the revised manuscript.

Comments 7: It is recommended to mention the challenges that may have the implementation of hydrogels in various medical applications.

Response: We thank the reviewer’s comment. We have supplemented the challenges that may have during the practical applications of wearable hydrogel sensors in various medical applications in the conclusion and future prospects part of the revised manuscript.

Round 2

Reviewer 4 Report

Comments and Suggestions for Authors

The authors made significant changes to the manuscript, so I recommend it for this journal.